# EAPO: Enhancing Policy Optimization with On-Demand Expert Assistance

**Siyao Song** [1 2 3 *]  **Cong Ma** [3 †]  **Zhihao Cheng** [3]  **Shiye Lei** [3]  **Minghao Li** [3]  **Ying Zeng** [3]  **Huaixiao Tou** [3]  **Kai Jia** [3]

## Abstract

Large language models (LLMs) have recently advanced in reasoning when optimized with reinforcement learning (RL) under verifiable rewards. Existing methods primarily rely on outcome-based supervision to strengthen internal LLM reasoning, often leading to inefficient exploration and sparse rewards. To mitigate this issue, we propose *Expert-Assisted Policy Optimization* (EAPO), a novel RL framework that enhances exploration by incorporating multi-turn interactions with external experts during training. Unlike prior methods, where policies reason in isolation, EAPO incentivizes the policy to adaptively determine when and how to consult experts, yielding richer reward signals and more reliable reasoning trajectories. External assistance ultimately internalizes expert knowledge into the policy model, amplifying the model's inherent reasoning capabilities. During evaluation, the policy model has been well-optimized to solve questions independently, producing improved reasoning paths and more accurate solutions. On AIME 2024/2025 and AIMO 2025, EAPO consistently outperforms expert-assisted, expert-distilled, and RL baselines, averaging a 5-point gain over self-exploration RL, and also generalizes to non-math benchmarks, including HumanEval, HLE, GPQA, MMLU, EvalPlus, HotpotQA, and SimpleQA.

## 1. Introduction

Large reasoning models (LRMs) have recently achieved remarkable progress in areas such as mathematics and logical reasoning (Chen et al., 2026b). However, guiding these models to produce accurate multi-step reasoning chains remains a major challenge, as it involves broad search spaces, sparse feedback, and unstable optimization process (Christiano et al., 2017; Gou et al., 2023). Existing post-training frameworks typically optimize models with supervised fine-tuning (SFT) and subsequently refine models through reinforcement learning from human feedback (RLHF) to better align model outputs with human preferences (Ouyang et al., 2022). Although effective in various applications, RLHF methods suffer from reward hacking, over-optimization on proxy objectives, and instability during training (Gao et al., 2023; Casper et al., 2023).

Reasoning models are particularly sensitive to these challenges, often manifesting as high exploration cost and unstable training dynamics when tasks require long-horizon reasoning. A growing body of work proposes more reliable reward signals to mitigate these issues. For instance, verifiable rewards provide objective supervision that reduces reward hacking (Lambert et al., 2024). Rita et al. (2024) calibrate rewards with demonstrations to prevent over-optimization, while Barj & Sautory (2024) leverage LLM-based feedback to address objective mis-generalization. Despite these advances in reward design, most existing RL algorithms still rely on the policy to independently explore the reasoning space and discover long chains of thought, a process that is often sample-inefficient and prone to suboptimal optimization. To improve reasoning performance at evaluation time, recent approaches instead scale inference via parallel sampling and aggregation over multiple candidate trajectories (Kumar et al., 2024), including self-consistency (Wang et al., 2022), Tree-of-Thoughts (Yao et al., 2023), Mixture-of-Agents (Wang et al., 2024), and LeaP (Luo et al., 2025). While effective in improving solution quality, these test-time scaling methods typically incur substantial communication overhead and implementation complexity, and crucially, they do not enhance the underlying capacity of the policy model itself.

To address these limitations, we propose **Expert-Assisted Policy Optimization** (EAPO), a novel reinforcement learning framework that explicitly integrates expert consultation into the policy's decision-making process (Figure 1). EAPO augments the agent's action space with a dedicated *consult-expert* action, allowing the policy to adaptively solicit ex-

---

*Work done during internship at ByteDance BandAI. †Corresponding authors. [1]State Key Laboratory of Multimodal Artificial Intelligence Systems (MAIS), Institute of Automation, Chinese Academy of Sciences, Beijing, China [2]School of Artificial Intelligence, University of Chinese Academy of Sciences, Beijing, China [3]ByteDance BandAI, Beijing, China. Correspondence to: Cong Ma <macong.13@bytedance.com>.

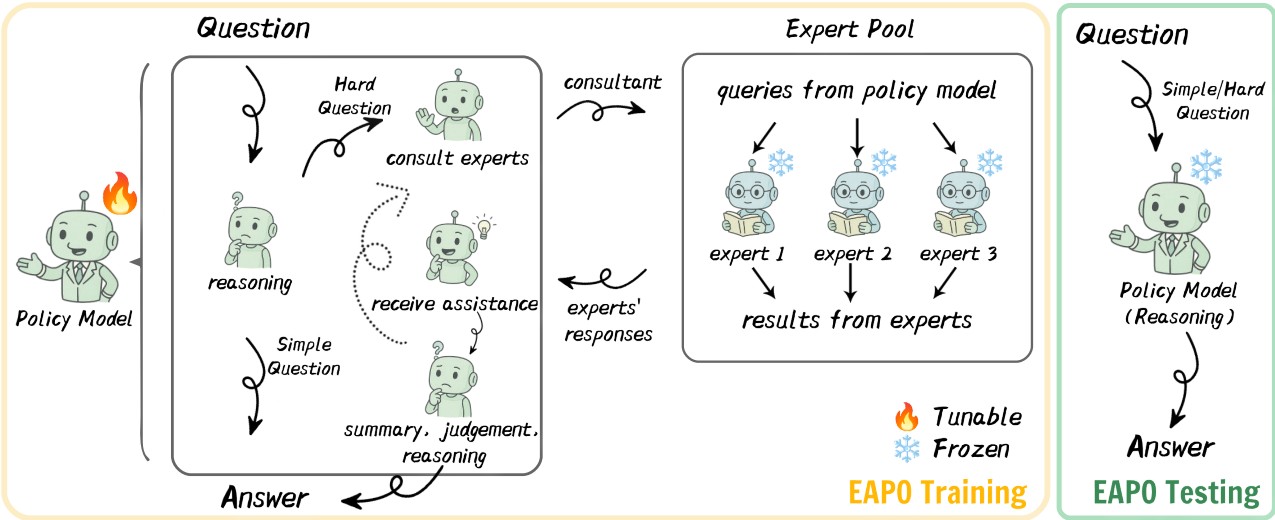

*Figure 1.* Framework of EAPO. During training, the policy model adaptively consults experts as assistants. While at test time, the model performs reasoning independently without external assistance.

ternal guidance when independent reasoning is insufficient. At each decision step, the policy can either generate reasoning tokens autonomously or invoke the *consult-expert* action to obtain auxiliary responses from an external expert pool, which serve as informative prompts that guide subsequent reasoning. Crucially, expert consultation is treated as a learnable action and optimized jointly with standard reasoning actions under the same RL objective. This design enables the agent to balance independent reasoning and expert assistance, while discouraging over-reliance on external experts during optimization.

During RL optimization, the model is progressively encouraged to rely on its own reasoning ability, ensuring that consultation occurs on demand only when the agent encounters difficult questions or ambiguous reasoning paths. With a curriculum-like optimization design, the agent is allowed to consult experts freely without in the early stages of training, providing richer assistance compared to the sparse feedback from final verifiable rewards. Experimental observations show that the policy model gradually learns to consult experts only for challenging questions, rather than relying on assistance for all cases. Moreover, responses from experts help the policy model to take deeper reasoning by consolidating multi-perspective insights. This design significantly improves the agent's advanced reasoning strategies, such as summarization, judgment, and reasoning, which are rarely observed in self-exploratory models. To further strengthen the model's own capability, a penalty term on consultation is introduced. As training progresses, the model is incentivized to consult less frequently, internalizing the consulted reasoning strategies into its own reasoning process. With sufficient training, EAPO effectively internalizes expert knowledge into the policy model, eliminating the need for consultation

during evaluation.

In our experiments, we comprehensively evaluate three datasets on mathematical reasoning, including AIME 2024, AIME 2025 and AIMO 2025. Compared with various strong baselines, such as expert-assisted workflow, distillation from experts and self-exploratory RL models, EAPO gains consistent improvements on all these benchmarks with a significant improvement of 5 points over self-exploratory RL on average evaluation results. Furthermore, EAPO yields lower test-set variance, indicating more stable and consistent reasoning. Importantly, EAPO is not limited to mathematical reasoning: evaluations on non-mathematical benchmarks demonstrate its strong generalization beyond the training domain. Our contributions are summarized as follows:

- **Expert-Assisted Optimization:** Expert consultation is designed as a learnable action within the policy, enabling the model to consult assistance on demand during training. This mechanism guides the reasoning trajectory throughout exploration and improves optimization efficiency compared to supervision based solely on final verifiable rewards.

- **Assistance Internalization:** EAPO progressively reduces the reliance on expert assistance throughout training, encouraging the model to internalize expert knowledge into its own reasoning process. During evaluation, the policy model responds entirely independently, yet still demonstrates strong reasoning performance.

- **Enhanced Reasoning Performance:** Extensive experiments on AIME 2024/2025 and AIMO 2025 show that EAPO improves accuracy and stability, and gener-

alizes to diverse non-math benchmarks including HumanEval, HLE, GPQA, MMLU, EvalPlus, HotpotQA, SimpleQA. Ablations further confirm consistent gains with increased expert parallelism and model scale.

## 2. Expert-Assisted Policy Optimization

### 2.1. Problem Formalization

We consider a complex reasoning task with input $x$ and ground truth answer $g$. A reasoning process can be formalized as a trajectory

$$H_T = \{(\tau_1, \alpha_1, o_1), \ldots, (\tau_t, \alpha_t, o_t), \ldots, (\tau_T, \alpha_T, o_T)\},$$

where $\tau_t$ represents the intermediate reasoning chain, $\alpha_t$ is the policy action that determines whether and how to consult the expert or produce the final answer, and $o_t$ denotes the assistance from experts at step $t$. $T$ represents the final step of the reasoning trajectory. If the policy calls the expert, then $o_t$ corresponds to the expert's feedback; otherwise, $o_t = \varnothing$. Given the current trajectory $H_{t-1} = \{(\tau_i, \alpha_i, o_i)\}_{i=1}^{t-1}$, the policy model $\pi_\theta$, parameterized by $\theta$, generates $\tau_t$ and $\alpha_t$ at step $t$ as

$$\pi_\theta(\tau_t, \alpha_t \mid H_{t-1}) = \pi_\theta^\tau(\tau_t \mid H_{t-1}) \cdot \pi_\theta^\alpha(\alpha_t \mid H_{t-1}, \tau_t),$$

where $\pi_\theta^\tau$ and $\pi_\theta^\alpha$ denote the probability of intermediate reasoning chain and the consultation decision, respectively. During testing, $\alpha$ is fixed to prevent expert consultation, ensuring that the policy model performs reasoning independently without expert assistance. Therefore, given an input $x$, the probability of generating the entire trajectory can be computed as

$$\pi_\theta(H_T \mid x) = \prod_{t=0}^{T-1} \pi_\theta^\tau(\tau_{t+1} \mid H_t)$$
$$\cdot \pi_\theta^\alpha(\alpha_{t+1} \mid H_t, \tau_{t+1}) p(o_{t+1} \mid \alpha_{t+1}).$$

where $H_0 = x$ denotes the initial input, and $p(o_{t+1} \mid \alpha_{t+1})$ is the probability of assistance returned by experts.

Once the full trajectory $H_T$ is generated by $\pi_\theta$, the output answer $\hat{y}$ is extracted from the last reasoning step using a pre-defined extraction rule, *i.e.*, $\hat{y} = \mathcal{E}(H_T)$. The reward function follows an end-to-end verifiable reward design. Given $\hat{y}$ and the ground truth be $g$, the reward is obtained as

$$R = \begin{cases} \text{f1}(\hat{y}, g) & \text{if f1 score is not } 0, \\ 0.1 & \text{if f1 score is } 0 \text{ with correct format}, \\ 0 & \text{otherwise} \end{cases}$$

Given the training dataset $\mathcal{D}$ consisting of $(x, g)$ pairs, the objective of EAPO can be defined as

$$\max_\theta \mathcal{J}_{\text{EAPO}} = \mathbb{E}_{(x,g) \sim \mathcal{D}, H_T \sim \pi_\theta(\cdot \mid x)} [R(\mathcal{E}(H_T), g)].$$

The key contribution of EAPO lies in introducing an expert assistant signal into the training process, which effectively accelerates and stabilizes optimization. At test time, the model is capable of reasoning independently while retaining the expert knowledge acquired during training, thereby bridging the benefits of expert-guided training and autonomous inference.

### 2.2. Mechanistic Intuition Behind EAPO

**Alleviating Sparse Rewards** EAPO enables the policy model to consult expert for assistance at critical steps during training, supplying cues of problem-solving from the external experts. In long-horizon reasoning tasks, merely relying on unguided exploration typically yields few positively rewarded samples. By incorporating external expert assistance, EAPO increases the likelihood of producing partially correct intermediate states and ultimately correct answers, densifying the reward signal, raising the proportion of effective trajectories, and consequently reducing inefficient search. This effect hinges on two assumptions: (i) the expert knowledge is valid for the current task and aligned with the objective, and (ii) the policy model can extract useful features from the assistance and transform them into better decisions, such as selecting more accurate descent directions or shrinking the search space. Under these conditions, EAPO acts as an implicit curriculum, guiding the model more rapidly into regions with high reward and establishing a solid foundation for subsequent incorporation of further external assistance or self-improvement.

**Information Gain** EAPO can be viewed as a lightweight mechanism of information injection: expert assistance acts as external signals that augment the historical context, enabling the policy model to select actions under richer conditioning and thereby tending to reduce decision uncertainty. Although this assistance does not constitute explicit process supervision, it provides local constraints and priors that are relevant to the goal. The expert assistance intuitively aligns the direction of update more closely with the reward structure and lowers the variance of gradient estimates. From an optimization perspective, expert queries bias exploration toward reward-consistent trajectories. As a result, selectively injected expert information helps the policy focus on reward-relevant intermediate decisions, internalize reusable heuristics, and achieve better performance with fewer training interactions. These effects are supported by our experiments and case studies.

### 2.3. End-to-End Training for Scalable Multi-Expert Parallel Reasoning

We model the action of asking experts for help as an explicit action available to the agent during training for solving complex tasks. At each step, the policy model $M_{\text{policy}}$

*Figure 2.* Three rollout paradigms on the training process: (1) Self-resolution (simple): Internal multi-step reasoning without expert assistance; (2) Direct Consultation (hard): Consult three experts directly about the problem and make a comprehensive decision through comparison. (3) Decomposition (complex): Break down into multiple sub-problems, consult experts, and combine the assistance to obtain the final answer. Examples of actual agents interacting with experts during training can be found in appendix F.

performs private reasoning, may query an expert $M_{\text{expert}}$ for assistance, integrates the returned information into its history, and then either continues the interaction or emits a final answer. Training unfolds in episodes with at most $T$ turns. At each turn, the policy model autonomously decides whether to consult the experts. Three rollout paradigms are observed during training, aligning with different task complexity and patterns of expert use, as shown in Figure 2: self-resolution for simple cases, direct consultation for hard cases, and decomposition for complex cases.

**Rollout Paradigms** For simple instances, the model solves the task through internal multi-step reasoning without consulting experts. The trajectory comprises private deliberation, iterative refinement, and termination with a final answer. Panel (1) in Figure 2 illustrates a representative self-resolution trajectory. It is worth noting that as training progresses, the model likewise adopts this paradigm whenever it has sufficient competence to handle the problem independently.

For harder instances, the model queries three experts on the original problem statement, obtains three independent answers to the same question, and aggregates them via comparison and reconciliation. This strategy improves robustness to single-path errors and reduces reliance on brittle

inference chains. Panel (2) of Figure 2 illustrates a typical trajectory involving direct expert consultation and answer aggregation.

For complex instances, the model first decomposes the task into multiple subproblems, consults experts on each component, and then aggregates the returned evidence into a coherent solution. This paradigm increases coverage, enables error localization, and promotes modular reasoning patterns that transfer across tasks. Panel (3) in Figure 2 presents decomposition, targeted consultations, and evidence fusion. Examples for the three paradigms mentioned above are shown in the appendix L.

**Parallel Evidence Gathering with Test-Time Alignment**
To broaden information coverage within a single turn while keeping computation manageable, we instantiate up to $K$ replicas of the expert model $\{ M_{\text{expert}}^{(i)} \}_{i=1}^{K}$. At turn $t$, $M_{\text{policy}}$ first reasons over the history $H_{t-1}$, then may select a concurrency level $C_t$ with $0 \leq C_t \leq K$, constructs structured queries, and dispatches them in parallel:

$$\mathcal{Q}_t = \{ q_{t,i} \}_{i=1}^{C_t}, \; r_{t,i} = M_{\text{expert}}^{(i)}(q_{t,i}), \; o_t = \{ r_{t,i} \}_{i=1}^{C_t},$$

where $q_{t,i}$ denotes the query sent to the $i$-th expert at itera-

*Table 1.* Evaluation on verifiable tasks. For the RL method, we train the model on DAPO-MATH, while other methods are implemented via offline workflows. We evaluate on AIME 2024, AIME 2025, and AIMO 2025. Specifically, **Pass@32** measures solution accuracy (the higher the better), while **Var** quantifies solution stability (the lower the better). We find that the EAPO consistently outperforms all comparison baselines across all three benchmarks. Additionally, some other findings on offline workflows can be found in appendix C.

| Category | Method | AIME 2024 | | AIME 2025 | | AIMO 2025 | | Avg. | |
|---|---|---|---|---|---|---|---|---|---|
| | | Pass@32↑ | Var↓ | Pass@32↑ | Var↓ | Pass@32↑ | Var↓ | Pass@32↑ | Var↓ |
| Base Model | Base Model | 51.73 | 0.1112 | 37.19 | 0.0580 | 38.68 | 0.1150 | 42.53 | 0.0947 |
| Offline Workflow | Self Collaboration | 50.27 | 0.2171 | 35.53 | 0.2004 | 39.68 | 0.2041 | 41.83 | 0.2072 |
| | Expert-Assisted Workflow | 58.07 | 0.2137 | 43.16 | 0.1978 | 46.94 | 0.2003 | 49.39 | 0.2039 |
| | LeaP (Luo et al., 2025) | 61.67 | 0.2547 | 39.27 | 0.2165 | 40.31 | **0.0244** | 47.08 | 0.1652 |
| Distillation (SFT) | Distillation (Full) | 43.54 | 0.0965 | 33.95 | 0.0927 | 28.44 | 0.1169 | 35.31 | 0.1020 |
| | Distillation (LoRA) | 51.90 | 0.1056 | 40.02 | 0.0618 | 37.81 | 0.1234 | 43.24 | 0.0969 |
| Online RL | Self-Exploratory RL | 67.16 | **0.0646** | 55.09 | 0.0625 | 55.22 | 0.0911 | 59.16 | 0.0727 |
| | **EAPO (Ours)** | **70.03** | 0.0673 | **58.01** | **0.0564** | **64.17** | 0.0693 | **64.07** | **0.0643** |

tion $t$, $r_{t,i}$ denotes the corresponding response returned by the $i$-th expert, and $o_t$ denotes the set of all expert responses collected at iteration $t$. The assistance $o_t$ are integrated to form $H_t$ and drive the next decision or produce the final answer. This design broadens the information gathered in a single turn, improves coverage of reasoning threads, and reduces reliance on any single path.

To promote agent autonomy and reduce reliance on experts, we anneal both the effective acceptance rate of expert calls and the per-episode turn budget during training. Let $s \in \mathbb{N}^+$ denote the global training step and $\rho_s \in (0,1]$ the probability that an expert response, once returned, is written into history. Early in training, a high acceptance rate accelerates the acquisition of useful patterns from expert feedback. As training progresses, we set $\rho_s = s^{-1}$, gradually suppressing expert access, encouraging internalization of the acquired knowledge, and shifting decision-making to the model's own policy. Concretely, for each expert response we draw $u \sim \mathrm{U}(0,1)$, where U denotes a sample from the standard uniform distribution. If $u \leq \rho_s$, the response is appended to the history, otherwise it is treated as unavailable and the policy model continues reasoning without external input. In parallel, the episode turn budget is reduced from its initial value toward the evaluation budget, which further promotes independent decision-making. The actual changes in the expert call rate during training can be seen in appendix B.

## 3. Experiments

### 3.1. Setup

**Models and Baselines** In this study, we adopt DeepSeek-R1-Distill-Qwen-7B (Guo et al., 2025) (hereafter,7B) as the $M_{\text{policy}}$ and reasoning backbone, and use QwQ-32B (Team, 2025) (hereafter,32B) as the $M_{\text{expert}}$ in heterogeneous collaboration settings. We compare our method against the following baselines: (1) Base Model corresponds to zero-shot

inference with 7B, which represents the initial capability. (2) Self Collaboration denotes an offline collaboration workflow in which the policy model is 7B and the experts are three replicas of 7B. (3) Expert-Assisted Workflow refers to a multi-agent workflow where 7B acts as the policy model while three replicas of 32B serve as experts. (4) LeaP (Luo et al., 2025) is a collaborative inference framework in which parallel paths conduct cross-path summarization and routing. (5) Full Parameter (Lv et al., 2023) and LoRA (Hu et al., 2022) distillation represent the full parameter and parameter efficient regimes. (6) Self-Exploratory RL (Yu et al., 2025) retains only outcome-driven reinforcement learning without external experts.

**Evaluation** We evaluate our method on three mathematical reasoning benchmarks: AIME 2024, AIME 2025, and AIMO 2025. Performance is measured using the mean and variance of accuracy, defined as follows:

- **Mean (Pass@32)**: For each problem, we independently sample 32 final answers and take the arithmetic mean of their F1 as the score for that problem; the dataset score is the mean over problems. Concretely, for problem $q$ with ground-truth answer $y^*$ and the $i$-th prediction $\hat{y}_i$, $\mathrm{Pass@32} = \frac{1}{32}\sum_{i=1}^{32}\mathrm{F1}(\hat{y}_i, y^*)$ and the overall dataset score is $\mathrm{F1} = \frac{1}{|Q|}\sum_{q \in Q}\mathrm{Pass@32}$.

- **Variance (Var)**: Owing to the instability of large model sampling, variance computation serves as an effective measure to assess stability. For each problem $q \in Q$ with ground-truth answer $y^*$, we draw $M = 32$ independent samples and compute $\mu_q = \frac{1}{M}\sum_{i=1}^{M}\mathrm{F1}(\hat{y}_i, y^*)$, where $\mu_q$ is the per-problem mean score (Pass@32). Aggregating over all $|Q|$ problems, we define $\overline{\mu} = \frac{1}{|Q|}\sum_{q \in Q}\mu_q$, $\mathrm{Var} = \frac{1}{|Q|}\sum_{q \in Q}(\mu_q - \overline{\mu})^2$.

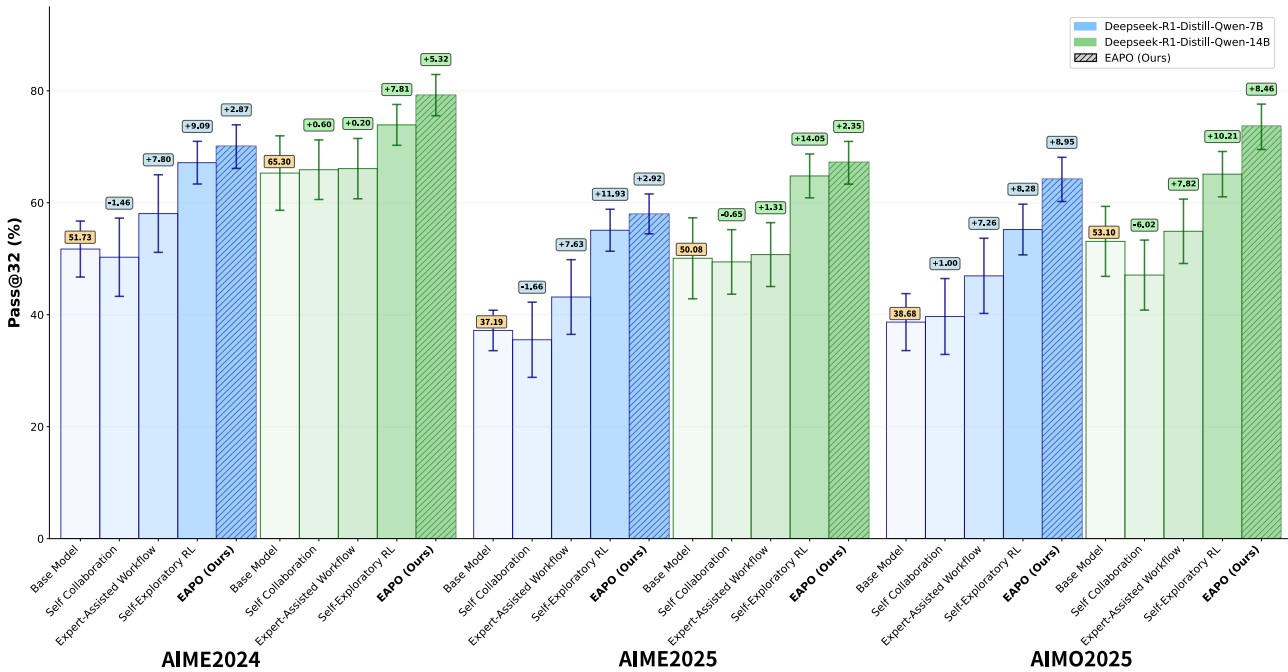

*Figure 3.* Policy model scaling under fixed expert (QwQ-32B). The model size of policy model is increased from 7B to 14B.

## 3.2. Experimental Results

Table 1 reports results on AIME2024, AIME2025, and AIMO2025 (Pass@32 and Var). From Table 1, our method attains the best Pass@32 on all three benchmarks, improving average F1 by 21.54% over the Base Model, 22.24% over Self Collaboration, 14.68% over Expert-Assisted Workflow, 16.99% over LeaP, 24.80% over the two distillation baselines, and 4.91% over self-exploratory RL. These findings further support the effectiveness of EAPO for policy optimization: by invoking expert assistance on demand, it enables more efficient exploration and strengthens LLM reasoning.

**Model Size and Policy Performance** To study the effect of the size of the policy model (Fig. 3), we replace the base model with DeepSeek-R1-distill-Qwen-14B and train from scratch under the same setup, keeping the expert fixed to QwQ-32B. Relative to the 7B model, the 14B model attains higher overall accuracy on the three datasets, but with diminishing marginal returns. Mechanistically, a larger size strengthens internal representations, enabling the model to resolve more difficult cases without expert consultation. However, as hard cases occupy a long-tail minority, incremental gains taper with scale.

**Interplay of Parallelism and Expert Capacity** We study two factors influencing training effectiveness, as summarized in Table 2: the parallelism of expert queries and the capacity of the expert model. For query parallelism, we

compare a sequential regime that issues one expert query per round with a parallel regime that allows up to three concurrent queries under the same overall interaction budget. We observe that parallel querying consistently yields higher solve rates. The most possible explanation is increased information coverage per episode: with a limited number of turns, issuing multiple queries simultaneously surfaces more relevant evidence before the budget expires, rescuing instances that sequential querying fails to resolve due to interaction limits.

For the model size of the expert, replacing a larger expert (QwQ-32B (Team, 2025)) with a smaller one (DeepSeek-R1-Distill-14B (Guo et al., 2025)) within the same end-to-end RL pipeline leads to a measurable performance drop. The smaller expert tends to provide less comprehensive and noisier guidance, which weakens the effective learning signal. Overall, parallel querying primarily improves exploration efficiency and robustness, whereas expert capacity governs the quality of injected information, both dimensions are necessary for stable, high-performing reasoning training.

**Homogeneous and heterogeneous expert pools.** We further ablate the structure of the expert pool (Table 3). When the experts are *homogeneous* to the policy (Deepseek-R1-distill-7B experts for a Deepseek-R1-distill-7B policy), EAPO's overall performance fluctuates around the pure-RL baseline. Together with our self-collaboration results shown in Table 1, this suggests that a homogeneous expert pool provides little information gain and may even amplify re-

*Table 2.* Effect of expert size and parallelism on reasoning performance. Parallel EAPO consistently outperforms self-exploratory RL and sequential EAPO. In addition, models trained with larger-size experts (32B) surpass those trained with smaller-size ones (14B).

| Method | Parallel | Expert Size | AIME 2024 | | AIME 2025 | | AIMO 2025 | | Avg. | |
|---|---|---|---|---|---|---|---|---|---|---|
| | | | Pass@32↑ | var↓ | Pass@32↑ | var↓ | Pass@32↑ | var↓ | Pass@32↑ | var↓ |
| Self-Exploratory RL | – | – | 67.16 | 0.0646 | 55.09 | 0.0625 | 55.22 | 0.0911 | 59.16 | 0.0727 |
| **EAPO (Ours)** | × | 32B | 68.31 | 0.0673 | 57.02 | 0.0675 | 60.03 | 0.0816 | 61.79 | 0.0721 |
| | √ | 14B | 68.93 | **0.0618** | 54.67 | 0.0693 | 61.06 | 0.0765 | 61.55 | 0.0692 |
| | √ | 32B | **70.03** | 0.0673 | **58.01** | **0.0564** | **64.17** | **0.0693** | **64.07** | **0.0643** |

*Table 3.* Ablation on homogeneous vs. heterogeneous expert pools in EAPO (pass@32 and var on AIME2024/2025 and AIMO2025 benchmarks). When the expert pool is homogeneous (Deepseek-R1-distill-7B experts for a Deepseek-R1-distill-7B policy), EAPO provides little to no gain over Self-Exploratory RL. In contrast, a heterogeneous expert pool (Llama-8B experts for a 7B policy) consistently improves performance, indicating that complementary capability distributions are important.

| Method | Expert | AIME2024 | | AIME2025 | | AIMO2025 | | Avg. | |
|---|---|---|---|---|---|---|---|---|---|
| | | pass@32↑ | var↓ | pass@32↑ | var↓ | pass@32↑ | var↓ | pass@32↑ | var↓ |
| Self-Exploratory RL | – | 67.16 | 0.0646 | 55.09 | 0.0625 | 55.22 | **0.0911** | 59.16 | **0.0727** |
| homogeneous EAPO | R1-distill-7B | 63.74 | 0.0613 | 54.73 | **0.0570** | 58.08 | 0.1084 | 58.85 | 0.0756 |
| heterogeneous EAPO | Llama-8B | **67.19** | **0.0604** | **55.92** | 0.0651 | **58.90** | 0.0927 | **60.66** | **0.0727** |

dundancy. In contrast, with a *heterogeneous* expert pool (Llama-8B experts for a Deepseek-R1-distill-7B policy), EAPO consistently outperforms pure RL on all three benchmarks. We attribute this to differences in capability distributions between Deepseek-R1-distill-7B and Llama-8B: EAPO can exploit this heterogeneity to realize complementary strengths between policy and experts, leading to more effective policy optimization. More generally, we posit that when the capability distributions of two models differ, they can learn from each other and become complementary, thereby improving overall optimization.

**Robustness and cross-domain generalization.** We further evaluate EAPO against expert-assisted workflows on a suite of non-mathematical benchmarks, including HumanEval (Chen, 2021), HLE (Phan et al., 2025), GPQA (Rein et al., 2024), MMLU (Hendrycks et al., 2020), EvalPlus (Liu et al., 2023), HotpotQA (Yang et al., 2018), and simpleQA (Wei et al., 2024). Notably, to isolate cross-domain transfer, we keep training strictly math-only, that EAPO is trained on the DAPO-MATH corpus without introducing any non-math data, and is then evaluated on non-math benchmarks. As shown in Table 4, despite this deliberate domain mismatch, EAPO delivers consistent and stable improvements over both the base model and the self-exploratory RL baseline across these diverse tasks. These results indicate that EAPO does not merely overfit to mathematical problem-solving patterns; instead, it internalizes broadly useful reasoning behaviors that transfer robustly beyond the training domain. Additionally, regarding the robustness of EAPO to expert capabilities under extreme conditions, see appendix E.

**Evolution of Response Length and Reasoning Style** Figure 4 presents the evolution of response length when training with EAPO compared to self-exploratory RL. In the early phase, EAPO responses are noticeably shorter. This reduction can be attributed to expert querying: by delegating segments of the reasoning process to external guidance, the model suppresses unnecessary exploratory text. As training progresses, the EAPO curve gradually exceeds self-exploratory RL. A plausible interpretation is that, after internalizing expert guidance, the model learns to construct more structured and explanatory chains of reasoning while improving performance.

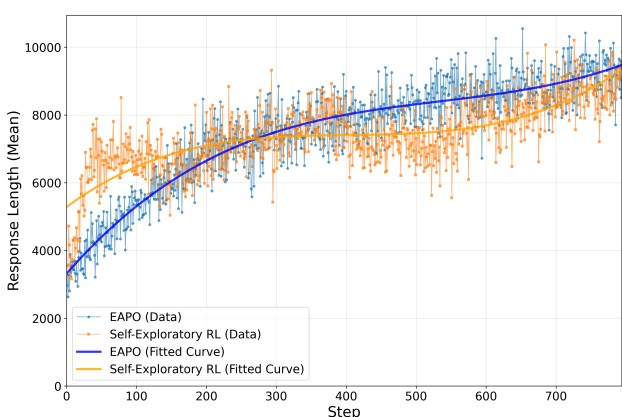

*Figure 4.* Dynamics of response length over training (compare EAPO with Self-Exploratory RL)

## 4. Related Work

**Expert-Guided Distillation** Expert-guided distillation aims to accelerate learning by coupling reinforcement ob-

*Table 4.* Generalization of EAPO from math-only training to non-mathematical benchmarks. EAPO consistently outperforms both the base model and a self-exploratory RL baseline across code and scientific tasks, indicating that expert-assisted policy optimization learned on math transfers effectively to other domains. Additionally, some findings on offline workflows can be found in appendix D.

| Method | HumanEval Code Generation | HLE Scientific Reasoning | GPQA Scientific Reasoning | MMLU Scientific Reasoning | EvalPlus Code Generation | HotpotQA Knowledge-Intensive Multi-Step | simpleQA Scientific Reasoning | Avg. Non-Mathematical Benchmark |
|---|---|---|---|---|---|---|---|---|
| Base Model | 86.79 | 7.00 | 45.92 | 48.16 | 83.72 | 12.86 | 6.70 | 41.59 |
| Self-Exploratory RL | 89.34 | 15.96 | 48.13 | 55.47 | 85.16 | 21.46 | 14.56 | 47.15 |
| **EAPO (Ours)** | **91.56** | **16.07** | **49.92** | **58.61** | **88.04** | **23.75** | **16.82** | **49.25** |

jectives with expert demonstrations. The core idea is that a student policy can consult an expert in uncertain states, imitate its behavior, and gradually reduce reliance as competence improves. Existing studies typically ground this idea in two forms. The first is *teacher-guided RL*, where a fixed teacher supervises the student and the student may eventually surpass the teacher in performance (Schmitt et al., 2018; Biré et al., 2024; Zhao et al., 2024). The second is *offline distillation*, where the learner regresses onto a static teacher using pre-collected trajectories (Rusu et al., 2016). More recent work emphasizes interactive imitation and selective querying, allowing expert signals to be injected at pivotal states to reduce distribution shift (Ross et al., 2011; Xie et al., 2022; Chi et al., 2020; Buck et al., 2017). Despite these advances, two challenges remain: (i) most methods assume persistent or externally scheduled reliance on experts, which risks over-dependence, and (ii) offline fitting cannot correct expert biases or adapt to environment-specific rewards. Our approach addresses both. We learn an explicit invocation policy that decides when to query the expert during on-policy interaction, so that supervision is obtained precisely where it is most useful. Crucially, the invocation rate is optimized end-to-end to converge to zero, ensuring that dependence vanishes as the student matures. This formulation both mitigates distribution shift and transforms expert guidance into a self-contained capability, yielding a policy model that is ultimately independent.

**Hierarchical Reinforcement Learning** Hierarchical RL (HRL) tackles long-horizon problems by decomposing control (Sutton et al., 1999; Bacon et al., 2017; Vezhnevets et al., 2017; Rashid et al., 2020; Yu et al., 2022; Chen et al., 2026a). Progress in this area has produced two dominant paradigms. One jointly trains options, termination conditions, and high-level controllers (Bacon et al., 2017), while another adopts a manager–executor structure where high-level goals condition low-level execution (Vezhnevets et al., 2017). These designs improve exploration but maintain permanent dependence on higher-level directives during both training and deployment. By contrast, our method treats the expert as a temporary external capability. Early in training, the expert is frequently invoked to stabilize learning, but as competence grows, invocations diminish until disappear. This yields two advantages: (i) we inherit the sample efficiency benefits of

hierarchical guidance in the early phase, and (ii) we guarantee that long-term execution becomes fully independent, avoiding the burden of permanent hierarchical control.

**Self Distillation and Social Learning** Self-distillation and related paradigms exploit a policy's own historical behaviors to bootstrap improvement. One line of work shows that a student distilled from its earlier versions can surpass the original policy under certain conditions (Furlanello et al., 2018). Self-imitation further reinforces validated behaviors by replaying high-return trajectories (Oh et al., 2018). Social learning demonstrates that complex strategies can be transferred between agents and eventually consolidated into individuals capable of independent execution (Ndousse et al., 2021; Vinyals et al., 2019). However, self-distillation often assumes unlimited replay of past trajectories and social learning presumes multiple agents remain available for transfer. Our work differs by a principled mechanism that ensures guidance accelerates early learning but disappears once competence is achieved, guaranteeing autonomy without needing auxiliary teachers indefinitely.

## 5. Conclusion

In this paper, we propose Expert-Assisted Policy Optimization (EAPO), a novel reinforcement learning framework designed to incorporate external assistance signals. The consultation of experts, a key component in EAPO, is formulated as a learnable action used solely during training, enabling the policy model to adaptively determine when and how to consult experts to improve exploration. Across three challenging mathematical reasoning benchmarks, EAPO outperforms both expert-assisted evaluation workflows and self-exploratory RL models. Beyond mathematics, EAPO also demonstrates robust cross-domain generalization on diverse non-math benchmarks, including HumanEval, HLE, GPQA, MMLU, EvalPlus, HotpotQA, and SimpleQA. Extensive experiments reveal that: 1) Incorporating expert assistance during RL training significantly improves policy optimization, yielding better reasoning paths and more accurate solution strategies. 2) Scaling expert assistance, through parallel signals or stronger expert models, consistently enhances performance, indicating that richer external signals during training can effectively strengthen optimiza-

tion. 3) Analyses across different policy model sizes show that expert consultation remains beneficial as student capacity increases. 4) EAPO internalizes broadly useful reasoning behaviors that transfer robustly beyond the training domain.

## Impact Statement

This paper presents work whose goal is to advance the field of machine learning, specifically reinforcement learning methods for improving the reasoning performance of large language models. The contributions of this work are methodological in nature and focus on training-time optimization under verifiable reward signals.

There are many potential societal consequences of advances in large language models and reinforcement learning, most of which are well established and broadly discussed in the existing literature. We do not believe that the methods introduced in this paper raise new ethical concerns beyond those commonly associated with prior work in this area, nor do they introduce novel deployment-time capabilities, application domains, or modes of interaction.

Accordingly, we do not feel that there are specific broader impacts that must be uniquely highlighted for this work.

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

## A. Implementation Details

The pre-trained policy model of DeepSeek-R1-Distill-Qwen-7B can be download via `https://huggingface.co/deepseek-ai/DeepSeek-R1-Distill-Qwen-7B` and DeepSeek-R1-Distill-Qwen-14B can be download via `https://huggingface.co/deepseek-ai/DeepSeek-R1-Distill-Qwen-14B`. The pre-trained expert model of QwQ-32B can be download via `https://huggingface.co/Qwen/QwQ-32B`. The training dataset DAPO-Math-17K is available at `https://huggingface.co/datasets/BytedTsinghua-SIA/DAPO-Math-17k`, and the evaluation datasets of AIME2024 and AIME2025 can be download on `https://huggingface.co/math-ai`, and AIMO2025 can be download on `https://www.kaggle.com/competitions/ai-mathematical-olympiad-progress-prize-2`. Table 5 lists the key hyperparameters used in self-exploratory RL and EAPO.

*Table 5.* Critical hyperparameters distilled from the training script.

| Hyperparameter | Value | Description |
|---|---|---|
| Max prompt length | 2048 | Input context window for prompts |
| Max response length | 20480 | Maximum generated tokens per sample |
| Train batch size | 32 | Number of prompts per training step |
| Gen batch size | 96 | Number of prompts per sampling batch |
| Learning rate | 1e-6 | Optimization step size |
| LR warmup steps | 10 | Steps to ramp LR up to target |
| Clip Ratio (Low) | 0.20 | Policy update stability window |
| Clip Ratio (High) | 0.28 | Policy update stability window |
| Temperature | 1.0 | Sampling stochasticity |
| Top-p | 1.0 | Nucleus sampling cutoff |
| Top-k | −1 | vLLM backend (no explicit top-k limit) |
| Sequence parallel (Ulysses) | 4 | Long-sequence efficiency via partitioning |
| Tensor model parallel | 4 | Inference/model-parallel scaling |

## B. Dynamics of Call Rate and Capability Internalization

To quantify how much the policy relies on experts—and how this reliance evolves over the course of training—we track the expert call rate, as shown in Figure 5. We observe a clear two-stage pattern. In the early phase, the policy invokes experts frequently to obtain cues for solution construction. As training progresses, the call rate decreases substantially, indicating that the policy is progressively internalizing useful expert behaviors and becoming less dependent on external guidance. Importantly, despite a large reduction in call frequency, final accuracy remains stable. Together, these trends evidence the joint effects of capability internalization and gating calibration.

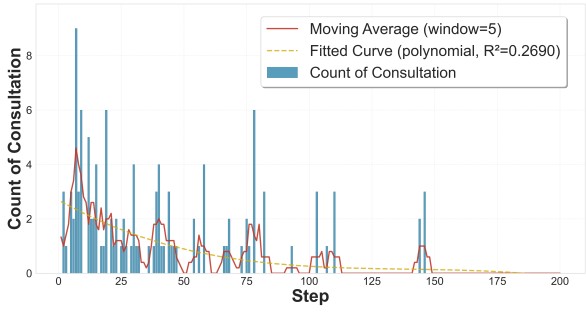
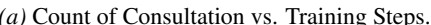
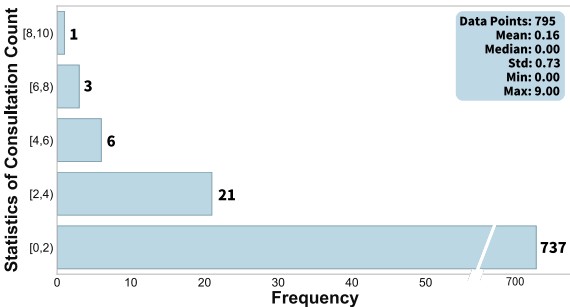

(a) Count of Consultation vs. Training Steps.    (b) Statistics of Consultation Count.

*Figure 5.* Rate of expert consultation during training. Early training relies heavily on experts; As capabilities are internalized and gates are calibrated, consultation rate of expert decreases, indicating the improvements of reasoning capacity.

## C. Conditional Expert-Assisted Workflows

To isolate the effect of expert heterogeneity in collaboration, we design an offline workflow (Table 6) and compare three settings: (1) **Single Model**, where the policy solves each problem independently; (2) **Self Collaboration**, where multiple

*Table 6.* Offline collaboration strategies compared with single-agent baselines. Heterogeneous expert-assisted workflow yields the largest gains for small models and offers minimal marginal benefit for large ones; self-collaboration is inconsistent and can degrade due to redundancy.

| Method | Policy Model | Experts | AIME2024 | AIME2025 | AIMO2025 | Average |
|---|---|---|---|---|---|---|
| Single Model | R1-7B | – | 51.73 | 37.19 | 38.68 | 42.53 |
| | R1-14B | – | 65.30 | 50.08 | 53.10 | 56.16 |
| | QwQ-32B | – | 78.25 | 60.81 | 71.52 | 70.19 |
| Self Collaboration | R1-7B | 7B+7B+7B | 50.27 | 35.53 | 39.68 | 41.83 |
| | R1-14B | 14B+14B+14B | **65.90** | **49.43** | 47.08 | 54.14 |
| | QwQ-32B | 32B+32B+32B | 78.07 | 64.07 | 73.95 | 72.03 |
| Expert-Assisted Workflow | R1-7B | 7B+14B+32B | **55.44** | **42.81** | **46.18** | **48.14** |
| | R1-14B | 7B+14B+32B | 64.65 | 48.65 | **53.98** | 55.76 |
| | QwQ-32B | 7B+14B+32B | **78.37** | **64.63** | **73.95** | **72.32** |

instances of the same policy generate parallel solutions that are then aggregated; and (3) **Expert-Assisted Workflow**, where the policy is provided with responses from external experts. The results reveal both the potential benefits and the practical limits of collaboration. In particular, self collaboration does not yield consistent improvements and can even degrade performance, suggesting that more trajectories alone is insufficient when they are produced by near-identical policies.

We attribute this failure mode to a concrete mechanism: redundancy amplification. Because self-collaborating agents share highly similar priors and reasoning patterns, their responses tend to be strongly correlated, often repeating the same intermediate steps, mistakes, or blind spots with minor surface-level variations.

By contrast, heterogeneous collaboration is most beneficial when policy model is small. Mutual-information gain is probably a driver that diverse experts contribute complementary priors and coverage. As policy model grows stronger, these benefits taper and can reverse, as weaker experts may introduce noisy or contradictory rationales. In summary, heterogeneous experts help chiefly by increasing usable information, whereas model size of experts governs response quality.

## D. Robustness and cross-domain generalization of offline workflow

*Table 7.* Offline workflow evaluation on non-mathematical benchmarks. Results show that expert assistance consistently improves over the single-agent baseline, indicating that expert-augmented workflows remain beneficial beyond mathematical reasoning.

| Method | Policy Model | Experts | HumanEval | HLE | GPQA | MMLU | EvalPlus | HotpotQA | simpleQA |
|---|---|---|---|---|---|---|---|---|---|
| Single Model | R1-7B | – | 86.79 | 7.00 | 45.92 | 48.16 | 83.72 | 12.86 | 6.70 |
| | R1-14B | – | 93.90 | 7.47 | 56.14 | 56.82 | 94.57 | 30.08 | 8.95 |
| | QwQ-32B | – | 59.96 | 8.34 | 59.67 | 67.03 | 60.61 | 34.09 | 12.31 |
| Expert-Assisted Workflow | R1-7B | 7B+7B+7B | **84.76** | 6.69 | 44.23 | 45.01 | 80.49 | 11.93 | 5.74 |
| | R1-7B | 14B+14B+14B | 83.54 | 7.27 | 46.06 | 52.59 | **80.91** | 27.85 | 8.27 |
| | R1-7B | 32B+32B+32B | 66.46 | **7.29** | **48.18** | **54.96** | 65.15 | **29.45** | **11.13** |

We construct offline collaboration workflows to disentangle the effect of collaboration structure, and compare self-collaboration against a *heterogeneous* Expert-Assisted Workflow on a suite of non-mathematical benchmarks. As shown in Table 7, incorporating external expert models yields consistent improvements over the single-agent baseline on most tasks, confirming that expert-augmented collaboration remains beneficial beyond mathematical reasoning. In contrast, self-collaboration provides limited gains, echoing our earlier findings that aggregating multiple highly correlated responses often introduces redundancy rather than complementary information.

An important nuance arises on code-centric benchmarks. In the last row (using QwQ-32B experts), the workflow does not improve HumanEval and EvalPlus; this is expected because QwQ-32B is itself relatively weak on these coding tasks, so its suggestions can be low-quality and may even distract the policy model, leading to a net performance drop. Nevertheless, the same setting still brings clear gains on the other five benchmarks (HLE, GPQA, MMLU, HotpotQA, and SimpleQA),

suggesting that expert assistance is most effective when the expert's capability profile matches or exceeds the target task requirements. Finally, while Table 7 isolates the effect of offline expert-assisted collaboration, Table 4 further demonstrates that our full method, EAPO, achieves improvements across *all seven* non-math benchmarks, indicating that expert signals are effectively internalized during training rather than merely providing test-time scaffolding.

## E. Noisy and meaningless experts: robustness in extreme settings

To further evaluate the robustness of EAPO under noisy or low-quality experts, we design two additional experiments:

**High-error experts.**    For each problem in AIME2024/2025 and AIMO2025, we build an "error expert" by selecting 3 incorrect trajectories from 32 QwQ-32B samples (if fewer than 3 are incorrect, we sample from all 32). This yields error rates above 50% (53.33% for AIME2024, 63.33% for AIME2025 and AIMO2025). We then evaluate DeepSeek-R1-distill-Qwen-7B on this collaboration dataset and compare it against an Expert-Assisted Workflow built from 3 randomly sampled trajectories.

**Completely meaningless guidance.**    In constructing collaboration trajectories, we replace all expert messages with fixed, content-free text (e.g., "I cannot offer you any assistance; please resolve the issue yourself."), simulating experts that are essentially uninformative.

*Table 8.* Comparison of sampling and guidance variants on AIME2024/2025 and AIMO2025 (pass@32 and var). Fully incorrect or uninformative guidance degrades performance, but the resulting bias remains bounded. The random-sampling strategy used by EAPO achieves the best overall results, improving the average pass@32 by ~7 points over error sampling and meaningless guidance, indicating that the guidance produced by EAPO is effective.

| Method | AIME2024 | | AIME2025 | | AIMO2025 | | Average | |
|---|---|---|---|---|---|---|---|---|
| | pass@32↑ | var↓ | pass@32↑ | var↓ | pass@32↑ | var↓ | pass@32↑ | var↓ |
| Error sampling | 53.73 | 0.2025 | 37.54 | 0.2157 | 37.18 | 0.2076 | 42.82 | 0.2086 |
| Meaningless guidance | 49.50 | **0.1906** | 40.05 | 0.2013 | 39.21 | **0.1940** | 42.92 | **0.1953** |
| **Random sampling (Settings in EAPO)** | **58.07** | 0.2137 | **43.16** | **0.1978** | **46.94** | 0.2003 | **49.39** | 0.2039 |

These settings simulate scenarios where the expert provides only weak or even purely noisy signals, and allow us to examine the stability and failure modes of EAPO under low-quality experts. Empirically, both fully incorrect and uninformative guidance lead to a consistent drop in pass@32 across AIME2024/2025 and AIMO2025, confirming that low-quality expert signals can misdirect the policy. However, the degradation remains bounded and the variance stays in a comparable range, suggesting that EAPO does not catastrophically overfit to noisy advice. In contrast, the random-sampling strategy used by EAPO achieves the strongest overall performance, improving the average pass@32 by roughly 7 points over error sampling and meaningless guidance, which indicates that the guidance exploited by EAPO is informative rather than merely adding noise.

## F. Interaction process during training

This section provides a concrete illustration of how the agent interacts with experts during training. The example shows a typical episode: the model first performs private deliberation and decomposes the task into sub problems, then dispatches structured queries to multiple experts in parallel, collects their replies, and integrates the returned evidence into the evolving history. When uncertainty remains, the model issues follow up checks before synthesizing the final answer.

The interaction trace also clarifies two important design choices. First, expert queries and returns are serialized in a simple, reproducible schema that makes the control flow explicit and auditable. Second, the policy model is encouraged to use experts not only for initial exploration but also for evidence confirmation. Together, these choices enable broad information acquisition within a turn while maintaining a disciplined loop of reasoning, consultation, and consolidation.

Model-Expert Loop

```
Model think:
<think>I decompose the problem into two subproblems.</think>
```

```
Model calls:
<agent_calls>[
  {"expert_id":1,"input_parameters":{"query":"Sub-question A"}},
  {"expert_id":2,"input_parameters":{"query":"Sub-question B"}}
]</agent_calls>

Model receives:
<agent_returns>[
  {"expert_id":1,"result":"Answer A."},
  {"expert_id":2,"result":"Answer B."}
]</agent_returns>

Model think:
<think>Let me confirm whether my previous reasoning and the answer given by the expert are correct.</
    think>

Model calls:
<agent_calls>[
  {"expert_id":1,"input_parameters":{"query":"Check-query A"}},
  {"expert_id":2,"input_parameters":{"query":"Check-query B"}},
  {"expert_id":3,"input_parameters":{"query":"Check-query C"}}
]</agent_calls>

Model receives:
<agent_returns>[
  {"expert_id":1,"result":"Response A."},
  {"expert_id":2,"result":"Response B."},
  {"expert_id":3,"result":"Response C."}
]</agent_returns>

Model think:
<think>With answers A and B and responses A, B, and C confirming the answers and previous thinking, I
    got the final answer.</think>

Final:
Answer: ...
```

## G. Prompts during Training and Evaluation

The training prompt grants the policy model the ability to coordinate experts, specify queries in a structured format, and integrate returned assistance into its own reasoning.

The evaluation prompt is deliberately minimal and focuses the policy model on producing a single, final answer in a standardized format. This design suppresses auxiliary process signals, aligns behavior with the evaluation environment, and enables fair comparison across methods and datasets. In combination, the training and evaluation prompts support broad information acquisition during learning while encouraging autonomy resoning at test time.

**Prompts during Training**

```
You are a helpful assistant that can solve a given question step by step. To do so, you can consult with
    a panel of three experts. Each question you can ask experts 10 times totally.

You can ask one or more experts for their opinion. You can ask them the same question or different
    questions simultaneously. The experts will provide their answers, which you can use to form your
    final response. The agent call query and results are enclosed within <agent_calls> </agent_calls>
    and <agent_returns> </agent_returns> tags respectively.

Agent calls query should be formatted as a list of dictionaries.
For each agent call (an item in the list), you need to provide these keys:
```

```
        (1) expert_id (int): Required. The ID of the expert you want to consult. Can be 1, 2, or 3.
        (2) input_parameters (dict): Required. All input parameters for the expert, in dictionary format.

Here are the experts you can use:
Description:
This is a panel of three auxiliary intelligent agents (ID 1, 2, 3) that you can consult when you are
        unsure about the answer to a question. The agents' responses may not always be correct, but they
        can offer you alternative perspectives to help you think further.

Parameters:
        (1) query (str): Required. The query string describing the question you want to ask or discuss.
        (2) The answer from the agent, including thinking process.

The reasoning process is wrapped by <think></think>.
The final answer begins with "Answer: ".

For example:
User: What is the capital city of NomanLand and what is its main export?
Assistant:
<think>
    This question has two parts. I will ask two different experts to answer each part. I'll ask expert 1
      about the capital and expert 2 about the main export.
</think>
<agent_calls>
    [{"expert_id": 1, "input_parameters": {"query": "capital city of NomanLand"}},
     {"expert_id": 2, "input_parameters": {"query": "main export of NomanLand"}}]
</agent_calls>
<agent_returns>
    [{"expert_id": 1, "result": "The capital city of NomanLand is NomanCity."},
     {"expert_id": 2, "result": "The main export of NomanLand is widgets."}]
</agent_returns>
<think>
    I see. Expert 1 says the capital is NomanCity, and Expert 2 says the main export is widgets.
</think>
Answer: The capital of NomanLand is NomanCity and its main export is widgets.
```

**Prompts during Evaluation**

```
You are a helpful assistant.

You must give the final answer in the last line, in the format of "Answer: xxx", xxx represents the
      final answer.
```

## H. Additional Ablation on the Consultation Schedule

To further examine the effect of the consultation schedule, we conduct an additional ablation comparing our annealed consultation strategy with three alternatives: a budget-matched fixed-rate baseline, a slower decay schedule, and a faster decay schedule. The goal of this comparison is to distinguish the benefit of annealing from the effect of simply changing the total amount of expert consultation.

Specifically, we compare the following schedules: a fixed consultation rate $\rho_s = 0.02$, which is matched to the total consultation budget of our method; a slower decay schedule $\rho_s = s^{-1/2}$; a faster decay schedule $\rho_s = s^{-2}$; and our default annealed schedule $\rho_s = s^{-1}$. Results are reported in Table 9.

The results show that our annealed schedule consistently achieves the best performance across all three benchmarks. Compared with the budget-matched fixed-rate baseline, our method improves the average score from 59.65 to 64.07, yielding a gain of 4.42 points. Since the fixed-rate baseline uses a matched total consultation budget, this improvement cannot be explained solely by the total number of expert calls. Instead, the result suggests that the temporal allocation of

*Table 9.* Ablation study on different consultation schedules. The fixed-rate baseline is budget-matched to our annealed schedule. Our default schedule $\rho_s = s^{-1}$ achieves the best average performance across all three benchmarks.

| Schedule | AIME2024 | AIME2025 | AIMO2025 | Avg. |
|---|---|---|---|---|
| $\rho_s = 0.02$ (Budget-Matched) | 67.92 | 55.13 | 55.89 | 59.65 |
| $\rho_s = s^{-1/2}$ (Slow Delay) | 68.53 | 57.41 | 62.53 | 62.82 |
| $\rho_s = s^{-2}$ (Faster Delay) | 68.04 | 55.21 | 56.03 | 59.76 |
| $\rho_s = s^{-1}$ (Ours) | **70.03** | **58.01** | **64.17** | **64.07** |

expert consultation is important.

The comparison with the slower and faster decay schedules further supports this interpretation. The slower schedule $\rho_s = s^{-1/2}$ maintains expert consultation for longer and achieves stronger performance than the fixed-rate baseline, but still underperforms our default schedule. In contrast, the faster schedule $\rho_s = s^{-2}$ removes expert consultation more aggressively and leads to substantially weaker final performance. These results indicate that expert access is most useful during the early, unstable phase of training, while a gradual withdrawal of consultation encourages the model to develop a stronger standalone policy.

## I. Additional Ablation with Stronger Expert Models

To further evaluate whether the improvement from the Expert-Assisted Workflow depends on a specific expert model, we conduct an additional ablation using stronger expert models while keeping the policy model fixed. In all experiments, the policy model is fixed as R1-7B, and only the expert model is varied. The experiments follow the same Expert-Assisted Workflow setting as in the main paper, where the notation $(\cdot) \times 3$ denotes the use of three expert-assisted samples or consultations under the same workflow configuration.

The results are shown in Table 10. Compared with the single-model R1-7B baseline, all expert-assisted variants substantially improve performance across the three benchmarks. Moreover, replacing the original expert with stronger experts further improves the average score, suggesting that the benefit of expert assistance is not tied to a particular expert model and does not saturate at the original expert scale.

*Table 10.* Ablation study with stronger expert models while keeping the policy model fixed as R1-7B. Stronger experts consistently improve the final performance under the same Expert-Assisted Workflow setting.

| Method | Expert | AIME2024 | AIME2025 | AIMO2025 | Avg. |
|---|---|---|---|---|---|
| Single Model | – | 51.73 | 37.19 | 38.68 | 42.53 |
| Expert-Assisted Workflow | (Qwen3-235B-A22B)×3 | 60.72 | 49.58 | 47.54 | 52.61 |
| Expert-Assisted Workflow | (GPT-5)×3 | 61.57 | 52.96 | 49.07 | 54.53 |
| Expert-Assisted Workflow | (DeepSeek_V3.2)×3 | **61.89** | 51.24 | **50.52** | **54.55** |

The single-model R1-7B baseline achieves an average score of 42.53, while the expert-assisted variants achieve average scores between 52.61 and 54.55. This corresponds to an absolute improvement of 10.08–12.02 points over the standalone policy model. These results indicate that expert assistance provides substantial gains even when the policy model itself is unchanged.

Among the stronger experts, GPT-5 and DeepSeek_V3.2 achieve higher average performance than Qwen3-235B-A22B, with DeepSeek_V3.2 obtaining the best overall average score of 54.55. This suggests that stronger expert models can provide more effective assistance to the same policy model. Together with the comparisons using smaller experts in the main paper, these results further suggest an expert-size scaling effect: larger or stronger expert models tend to induce better final performance under the same Expert-Assisted Workflow framework.

Overall, this ablation supports two conclusions. First, the effectiveness of the Expert-Assisted Workflow is not specific to the original expert model used in the main experiments. Second, the improvement does not appear to saturate at the original expert scale, as stronger experts continue to yield additional gains while the policy model remains fixed.

## J. Additional Policy Backbone: Qwen3-8B

To examine whether the effectiveness of EAPO generalizes beyond the R1-series policy models used in the main experiments, we conduct an additional set of experiments using Qwen3-8B as the policy backbone. We compare four settings: the base Qwen3-8B model, the Expert-Assisted Workflow, self-exploratory RL, and EAPO. The results are reported in Table 11.

*Table 11.* Results on Qwen3-8B as an additional policy backbone. EAPO consistently achieves the best Pass@32 performance across all benchmarks and improves over the base model, the expert-assisted workflow, and self-exploratory RL.

| Method | AIME2024 | | AIME2025 | | AIMO2025 | | Avg. | |
|---|---|---|---|---|---|---|---|---|
| | Pass@32 | Var | Pass@32 | Var | Pass@32 | Var | Pass@32 | Var |
| Base Model | 40.84 | 0.1251 | 22.62 | 0.0812 | 23.57 | 0.1281 | 29.01 | 0.1114 |
| Expert-Assisted Workflow | 44.37 | 0.1960 | 28.16 | 0.2127 | 30.01 | 0.2224 | 34.18 | 0.2104 |
| Self-Exploratory RL | 50.21 | 0.0629 | 43.52 | 0.0634 | 42.59 | 0.0811 | 45.44 | 0.0691 |
| EAPO (Ours) | **53.84** | **0.0610** | **47.03** | **0.0578** | **49.08** | **0.0743** | **49.98** | **0.0644** |

The results show a consistent trend with the observations on R1-7B and R1-14B in the main experiments. EAPO achieves the best Pass@32 performance on all three benchmarks. In addition to the improvement in Pass@32, EAPO also achieves the lowest average variance among all compared methods. This indicates that the improvement is not merely due to increased exploration, but is accompanied by more stable final performance.

Overall, this ablation demonstrates that the benefit of EAPO is not specific to the R1-7B or R1-14B policy backbones. Instead, EAPO also provides clear gains when applied to Qwen3-8B, suggesting that the proposed training strategy generalizes to other recent reasoning-oriented policy models.

## K. vLLM Server

We deploy the expert model as an on-demand inference service through vLLM. The service is exposed to the outside world through a standardized HTTP/OpenAI interface, supports long-context reasoning, and can be efficiently scaled in a multi-GPU environment. The training end maps the query set constructed in each round into a request and sends it to the server. After the request is returned, it is written into the history, thus supporting the parallel training process from information acquisition to output decision.

**Inference Service through vLLM**

```
vllm serve ${MODEL_PATH} \
    --port ${PORT} \
    --host :: \
    --gpu-memory-utilization 0.8 \
    --max-model-len 32768 \
    --tensor-parallel-size 8 \
    --served-model-name expert \
    --trust-remote-code \
    --disable-log-requests
```

## L. Case Study

Below are three sets of cases corresponding to the three rollout paradigms in Section 2.3. All are derived from real interaction records during the training phase (minimized for readability). They illustrate three collaborative modes: self-resolution, direct consultation, and decomposition. In the first case, the model relies entirely on internal multiple step reasoning to arrive at an answer; in the second, the model consults multiple experts concurrently on the same problem, forming a comprehensive judgment through cross comparison; and in the third, the model first breaks a complex task into several sub-problems, poses targeted questions to the experts, and gradually integrates the evidence to ultimately synthesize a global solution.

## Self Resolution

User: Let $a, b, c$ be distinct numbers such that the equations $x^2 + ax + 1 = 0$ and $x^2 + bx + c = 0$ have a common real root, and the equations $x^2 + x + a = 0$ and $x^2 + cx + b = 0$ also have a common real root. Compute the sum $a + b + c$.

Assistant:
```
<think>
    Step 1. Let the common root of the first pair be r:
        r^2 + ar + 1 = 0,  r^2 + br + c = 0
        (a-b)r + (1-c) = 0
        r = (c-1)/(a-b).
    Step 2. Let the common root of the second pair be s:
        s^2 + s + a = 0,  s^2 + cs + b = 0
        (1-c)s + (a-b) = 0
        s = (b-a)/(1-c).
    Step 3. Combine the two:
        rs = ((c-1)/(a-b)) * ((b-a)/(1-c)) = 1
        s = 1/r.
    Step 4. Substitute s = 1/r into s^2 + s + a = 0:
        (1/r^2) + (1/r) + a = 0
        1 + r + a r^2 = 0.
        From r^2 + ar + 1 = 0
        r^2 = -ar - 1. Substituting:
        1 + r + a(-ar - 1) = 0
        r(1 - a^2) + (1 - a) = 0.
        If a != 1, then r = (a-1)/(1-a^2) = -1/(1+a).
    Step 5. Substitute r = -1/(1+a) back into r^2+ar+1=0:
        (1/(1+a)^2) - a/(1+a) + 1 = 0
        (1+a)^2 + 1 - a(1+a) = 0   a+2=0   a=-2.
    Step 6. Then r = -1/(1-2) = 1. Plugging into x^2+bx+c=0 gives 1+b+c=0
        b+c=-1.
    Step 7. Therefore a+b+c = -2 + (-1) = -3.
</think>
Answer:-3
```

## Direct Consultation

User: The integers from $1$ to $n$ are written, one on each of $n$ cards. The first player removes one card. Then the second player removes two cards with consecutive integers. After that, the first player removes three cards with consecutive integers. Finally, the second player removes four cards with consecutive integers. What is the smallest value of $n$ for which the second player can ensure that he completes both his moves?

Assistant:
```
<think>
    Provide three distinct approaches (interval-splitting, hitting-set of 4-blocks, and fragmentation/
    invariant), then synthesize them to obtain the minimal n.
</think>
<agent_calls>
    [
      {"expert_id": 1, "input_parameters": {"query": "Find the minimal n guaranteeing the second player
    can remove 2 consecutive and later 4 consecutive, regardless of the first player's moves."}},
      {"expert_id": 2, "input_parameters": {"query": "Recast the game as a hitting-set/covering problem
    on all 4-consecutive windows; derive threshold n."}},
      {"expert_id": 3, "input_parameters": {"query": "Use fragmentation invariants (remaining length = 8
    after three moves) to characterize when a 4-run must remain; produce a constructive strategy."}}
    ]
</agent_calls>
<agent_returns>
```

{"expert_id": 1, "result": "Interval-splitting method. After the first player removes a single card at position $p$, the remaining integers form two contiguous segments of lengths $A=p-1$ and $B=n-p$ (so $A+B=n-1$). The second player's 2-consecutive removal inserts a 2-gap inside one of these segments, splitting it into two parts whose lengths sum to (chosen segment) $-2$. If, before the first player's 3-consecutive removal, we can ensure at least two segments have length $\ge 4$, then no single 3-block can destroy both; hence a 4-block will survive for the final move. Sufficiency for $n=14$: Since $A+B=13$, either $\min(A,B)\le 3$, which forces $\max(A,B)\ge 10$, letting us split the larger segment into two parts both $\ge 4$ by removing two cards near its middle (since $10-2=8=4+4$); or else $\min(A,B)\ge 4$ and then $\max(A,B)\ge 9\ge 6$, so by removing two from the larger side we can keep one part $\ge 4$ while the other side already contributes another $\ge 4$. Thus after our move there are at least two $\ge 4$ segments; the opponent's 3-block can hit at most one of them, leaving a 4-run intact. Necessity: For $n=13$, pick $p=4$ so $(A,B)=(3,9)$. Whatever 2-consecutive the second removes, after that there is at most one segment of length $\ge 4$. The first player then deletes three consecutive from the length-9 segment, centered to split it into $(3,3)$, leaving no 4-run. Therefore $n$ must be at least $14$.", "status": "SUCCESS_AGENT"}
{"expert_id": 2, "result": "Hitting-set viewpoint on 4-windows. Consider the family of all 4-consecutive windows W_i = {i,i+1,i+2,i+3} for i=1,...,n-3. The second player's final success is equivalent to the existence of a window W_i fully contained in the remaining set after three moves (sizes removed: 1, then 2, then 3 -> total 6 removed). To prevent a 4-run, the six removed numbers must hit every W_i (i.e., form a 4-cover).\n\nWhen n=13, the first player can force such a 4-cover: remove {4} first, and no matter which two consecutive the second removes, the first can choose a 3-block (e.g., {6,7,8} in a typical case) so that every 4-window intersects one of the removed positions. Hence the second cannot guarantee a 4-run at the end. For n=14, the second can always choose his 2-consecutive so that there exist at least two disjoint 4-windows not simultaneously hittable by a single 3-block; thus after the opponent's move, at least one 4-window survives. This yields the threshold n=14.", "status": "SUCCESS_AGENT"}
{"expert_id": 3, "result": "Fragmentation invariant & constructive plan. After the first three moves, exactly 6 numbers are removed and 8 remain. To eliminate all 4-runs, the opponent must fragment the 8 survivors into pieces of maximum length <= 3 (e.g., 3+3+2 or 3+2+2+1). There are only three gaps available (sizes 1,2,3 from the three deletions), so the second player's objective in his first move is to place the 2-gap so that, regardless of where the later 3-gap is placed, the maximum surviving piece after that removal is >= 4.\n\nA constructive strategy at n=14: Observe the split (A,B) with A+B=13. If min(A,B) <= 3, then the other side has length >= 10; remove two near its middle to create two subsegments of lengths >= 4 each. If both sides are >= 4, then the larger side is >= 9; remove two within the larger side so that one subpiece remains >= 4 while the smaller original side (>= 4) is untouched. In both subcases, after the second player's move there are two disjoint pieces of length >= 4, and a single 3-block cannot destroy them both, ensuring a 4-run survives. For n=13, the adversarial split (3,9) defeats this plan, as any 2-gap yields at most one piece >= 4, which can be broken by a centered 3-removal into pieces of size <= 3.", "status": "SUCCESS_AGENT"}
</agent_returns>
<think>
    All three perspectives agree on the same threshold. The interval-splitting method formalizes a
    guaranteed way for the second player, at n=14, to leave two >=4 segments before the final 3-removal;
     a single 3-block cannot eliminate both. The hitting-set view explains why n=13 fails: the first
    player can force a 4-cover with six deletions. The fragmentation invariant reframes the goal as
    preventing all pieces from being of length <=3 and yields a simple constructive rule for where to
    place the 2-gap. Thus the smallest n for which the second player can ensure completing both moves
    is 14.
</think>
Answer: 14

---

## Decomposition

User: Find all positive integers $n$ for which all positive divisors of $n$ can be placed into the cells of a rectangular table under the following constraints:
- Each cell contains a distinct divisor.
- The sums of all rows are equal.
- The sums of all columns are equal.

```
Assistant:
<think>
    Decompose the task:
    (A) Structural necessities: if the divisors of n fill an m*k grid ($mk=\tau(n)$), then row/column
    sums must be $\sigma(n)/m$ and $\sigma(n)/k$ (integers), so $m \mid \sigma(n)$, $k \mid \sigma(n)$;
     also rule out $1\times k$ or $k\times 1$.
    (B) Exclude any $2\times k$ case for $n>1$ by showing equal column sum forces each column to be $(x
    ,\, n+1-x)$, which is impossible with distinct divisors.
    (C) Handle $\tau(n)$ odd ($n$ a perfect square): argue that $m,k\geq 3$ leads to a contradiction via
     extremal-sum bounds; conclude no solution for $n>1$.
</think>
<agent_calls>
    [
     {"expert_id": 1, "input_parameters": {"query": "Derive necessary divisibility/size constraints on
    m,k from the equal-row/column-sum conditions; handle 1*k and k*1 edge cases."}},
     {"expert_id": 2, "input_parameters": {"query": "Prove impossibility for any 2*k arrangement with
    all divisors when n>1 (pair-sum argument leading to S = n+1)."}},
     {"expert_id": 3, "input_parameters": {"query": "Treat $\tau(n)$ odd (n is a perfect square): show
    no m*k with m,k>=3 can work for n>1 using extremal averages/paired-divisor bounds; provide a
    consistency check."}}
    ]
</agent_calls>
<agent_returns>
    {"expert_id": 1, "result": "Let $\\tau(n)$ be the number of positive divisors and $\\sigma(n)$ their
     sum. If the divisors fill an $m\\times k$ rectangle, then $mk=\\tau(n)$, each row sums to $\\sigma
    (n)/m$, and each column sums to $\\sigma(n)/k$; hence $m\\mid\\sigma(n)$ and $k\\mid\\sigma(n)$. If
     $m=1$ (or $k=1$), then all column (row) sums must be equal across singleton columns (rows),
    forcing all entries equal---impossible with distinct divisors unless $\\tau(n)=1$, i.e., $n=1$. So
    for $n>1$ we must have $m,k\\ge 2$.", "status": "SUCCESS_AGENT"}
    {"expert_id": 2, "result": "Rule out $2\times k$ for $n>1$. In a $2\times k$ table with equal column
     sums, every column must consist of a pair whose sum is a constant $S$. Since the set is all
    divisors of $n$, one column contains $n$, so $S-n$ must be a positive divisor; thus $S>n$ and hence
     $S\geq n+1$. On the other hand, for any divisor $d$, we have $d + \tfrac{n}{d} \leq n+1$ with
    equality iff $\{d,\, n/d\}=\{1,\,n\}$. Therefore the only way to have every column sum equal to a
    common $S\geq n+1$ is to set $S=n+1$ and make every column the pair $(1,\,n)$. That is impossible
    once $k\geq 2$ because there is only one $1$ and one $n$. Hence no $2\times k$ arrangement exists
    for $n>1$ (and similarly no $k\times 2$).", "status": "SUCCESS_AGENT"}
    {"expert_id": 3, "result": "Treat $\tau(n)$ odd $\Rightarrow n$ is a perfect square and any
    rectangular factorization must have $m,k$ odd. The smallest nontrivial case is $3\times 3$. Let the
     divisors be $1=d_1<d_2<\cdots<d_9=n$. Column sums must all equal $S_c=\sigma(n)/3$. The column
    containing $n$ has sum at least $n+d_1+d_2=n+1+d_2$. Any column avoiding $n$ has sum at most $d_7+
    d_8+d'_9$ where $d'_9\leq d_8<n$; by the paired-divisor bound, the average of any pair $(d,\,n/d)$
    is $\leq (n+1)/2$, implying $\sigma(n)/\tau(n)\leq (n+1)/2$ and hence $S_c=\sigma(n)/3\leq 3(n+1)/2
    $. Comparing the column with $n$ versus one without forces $S_c\geq n+1+d_2$ and simultaneously
    $S_c\leq 3(n+1)/2$; for $n>1$ these bounds require each non-$n$ column to include an element $\geq
    (n+1)/2$. But there are at most two divisors $\geq (n+1)/2$ other than $n$ itself (namely $\lceil \
    sqrt{n}\rceil$ and its complement), while there are at least two non-$n$ columns in a $3\times 3$
    grid, and for larger odd $m,k$ the shortage worsens. Hence no $m,k\geq 3$ arrangement exists for $n
    >1$. This is consistent with computational checks for $n\leq 60$, which find no example beyond $n=1
    $.", "status": "SUCCESS_AGENT"}
</agent_returns>
<think>
    Synthesis: From (1) we reduce to $m,k\geq 2$ for $n>1$. From (2) any arrangement with $\min(m,k)=2$
    is impossible. Thus if $n>1$, we would need $m,k\geq 3$; but (3) shows this cannot happen when $\
    tau(n)$ is odd (squares), and even $\tau(n)$ already fails by (2). Therefore the only $n$ that
    works is $n=1$ ($\tau=1$; the $1\times 1$ table trivially satisfies all conditions).
</think>
Answer: 1
```

## M. Alternative Contact Email

If the corresponding author's email address listed on the title page becomes unavailable, correspondence regarding this paper may be directed to macong275262544@outlook.com.

