# OpenReview forum: "EAPO: Enhancing Policy Optimization with On-Demand Expert Assistance"
_ICML.cc/2026/Conference — ICML 2026 regular_

### Official Review · Reviewer_s4n5 · 2026-02-25

**Soundness:** 3
**Presentation:** 3
**Significance:** 2
**Originality:** 2
**Overall Recommendation:** 3
**Confidence:** 3

**Summary:**

The paper introduces Expert-Assisted Policy Optimization (EAPO), a reinforcement learning framework designed to enhance the reasoning capabilities of Large Language Models (LLMs). EAPO augments the policy model's action space with a "consult-expert" action, allowing it to dynamically query an external pool of experts (e.g., QwQ-32B) during the training phase to obtain guidance on complex reasoning tasks. To prevent over-reliance on the experts, the framework employs an annealing mechanism that gradually reduces the probability of incorporating expert feedback, forcing the policy to internalize the reasoning capabilities. The authors demonstrate that EAPO outperforms offline expert-assisted workflows and standard self-exploratory RL on mathematical reasoning benchmarks (AIME 2024/2025, AIMO 2025) and shows out-of-domain generalization.

**Compliance With Llm Reviewing Policy:**

Affirmed.

**Final Justification:**

I appreciate the substantial effort the authors have made to address my concerns. It is clear that they have worked hard on the rebuttal and solved most of my concerns. However, I still believe that memory usage remains the main bottleneck of the paper (10 times memory usage compared with baseline, it may seriously limited its scalability). Although I maintain this concern, I am willing to lower my confidence.

**Key Questions For Authors:**

1. Could you provide a quantitative comparison of the wall-clock training time, GPU hours between EAPO and the Self-Exploratory RL baseline? How do you address the massive generation latency introduced by waiting for parallel responses from three 32B experts during the RL rollout phase?
2. Could you provide an On-Policy Distillation baseline (as well as the GPU hours).
3. What happens to the policy's performance and consultation frequency if the hard-coded annealing schedule  is completely removed, and the model is purely regularized by a step penalty or KL penalty for consulting experts?

**Limitations:**

1. The effectiveness of EAPO is heavily bounded by the capability of external experts. As acknowledged by the authors in the code-centric benchmarks, if the expert model is weak in a specific domain, the consultation mechanism can distract the policy model and degrade performance.
2. The current experiments use the policy model at 14B. Scaling EAPO to train 30B+ or 70B+ policy models seems computationally unfeasible under standard academic or mid-tier industry compute budgets.

**Strengths And Weaknesses:**

Strength:
1. Framing expert consultation as a learnable action during the RL rollout phase is an interesting approach to mitigating the sparse reward problem in long-horizon reasoning tasks
2. The paper provides solid experimental results across challenging math benchmarks and includes out-of-domain evaluations demonstrating that the internalized capabilities transfer reasonably well.

Weakness:
1. The framework requires running a target policy model with multiple, and maybe larger models during the RL rollout phase. Rollout generation is already the primary bottleneck in LLM RL training. Multiple experts models for multi-turn interactions introduces severe memory constraints and massive communication latency, making this approach extremely difficult to scale.
2. Lack of Strong Baselines (On-Policy Distillation): The method inherently relies on stronger experts dynamically injecting knowledge into the policy model's context during exploration. This paradigm is conceptually very similar to On-Policy Distillation, where a teacher guides the student's on-policy generations.
3. The paper claims the model learns to adaptively consult experts, but the effective acceptance rate of expert calls is using a hard-coded method. This mechanism doesn't act as a learned gating parameter driven by the RL reward objective.

---

> ### Author Rebuttal · Authors · 2026-03-31
>
> **For W1 & Q1:**
>
> The training cost of EAPO relative to the self-exploratory RL baseline is reported in **Reviewer CQyp's W3/Q3**. The overhead of the three 32B experts arises only during training. Since the three experts are queried in parallel, the added rollout latency is determined by the slowest expert, rather than the sum of all three. In addition, we observe that EAPO converges faster during training, which partially offsets the extra computation. Overall, EAPO introduces a moderate increase in training-time overhead while improving inference-time efficiency. We will add training curve comparison to camera ready.
>
>
>
> &nbsp;
>
>
>
> **For W2 & Q2:**
>
> We thank the reviewer for this helpful suggestion and agree that on-policy distillation is a highly relevant baseline. We therefore implemented this baseline (22 hours on 8*8 H20 GPUs) and found that, while it is substantially stronger than off-policy distillation, it remains consistently below EAPO. This suggests that EAPO’s gain is not due to on-policy teacher imitation alone, but to the combination of expert interaction during exploration and RL-based policy optimization.
>
> |                                | AIME2024  | AIME2024   | AIME2025  | AIME2025   | AIMO2025  | AIMO2025   | Avg.      | Avg.       |
> | ------------------------------ | --------- | ---------- | --------- | ---------- | --------- | ---------- | --------- | ---------- |
> |                                | Pass@32   | Var        | Pass@32   | Var        | Pass@32   | Var        | Pass@32   | Var        |
> | Off-Policy Distillation (LoRA) | 51.90     | 0.1056     | 40.02     | 0.0618     | 37.81     | 0.1234     | 43.24     | 0.0969     |
> | On-Policy Distillation         | 63.43     | 0.0837     | 52.29     | 0.0729     | 54.47     | 0.1026     | 56.73     | 0.0864     |
> | EAPO (Ours)                    | **70.03** | **0.0673** | **58.01** | **0.0564** | **64.17** | **0.0693** | **64.07** | **0.0643** |
>
>
>
> &nbsp;
>
>
>
> **For W3 & Q3:**
>
> We thank the reviewer for this important point. We have added ablation comparing the annealed consultation schedule with a budget-matched fixed-rate baseline, together with slower and faster decay variants. Across all three benchmarks, the annealed schedule achieves the best final performance, outperforming both the fixed-rate baseline and the alternative decay schedules.
>
> The results are consistent with our intended interpretation: stronger expert access is most helpful in the early, unstable stage of training, while gradual withdrawal leads to a better final standalone policy than either keeping the consultation rate fixed or decaying it too slowly/too aggressively.
>
> Due to the limited rebuttal period, we have not yet completed the comparison where the hard-coded schedule is entirely removed and consultation is controlled only by a step penalty or a KL-style penalty. We agree this is an important direction and will include it in the revision.
>
> |                                    | AIME2024  | AIME2025  | AIMO2025  | Avg.      |
> | ---------------------------------- | --------- | --------- | --------- | --------- |
> | $$\rho _s=0.02$$  (Budget-Matched) | 67.92     | 55.13     | 55.89     | 59.65     |
> | $$\rho _s=s^{-1/2}$$  (Slow Delay) | 68.53     | 57.41     | 62.53     | 62.82     |
> | $$\rho _s=s^{-2}$$ (Faster Delay)  | 68.04     | 55.21     | 56.03     | 59.76     |
> | $$\rho _s=s^{-1}$$ (Ours)          | **70.03** | **58.01** | **64.17** | **64.07** |
>
>
>
> &nbsp;
>
>
>
> **For L1:**
>
> Thank you for this important question. EAPO is not strictly bounded by expert capability and does not require the expert to be uniformly stronger than the policy model. As you mentioned, although QwQ-32B underperforms R1-7B on code benchmarks such as HumanEval and EvalPlus (Appendix D, Table 7), EAPO still improves code-task performance in this setting (Table 4). This suggests that EAPO can benefit from complementary strengths between the expert and the policy, even when the expert is imperfect in a given domain. In addition, Appendix E demonstrates that when experts are frequently wrong or completely uninformative, performance drops, but the degradation remains bounded.
>
>
>
> &nbsp;
>
>
>
> **For L2:**
>
> We agree that directly scaling EAPO to 30B+ or 70B+ policy models would incur higher training cost under typical academic or mid-scale industry compute budgets. Our 7B and 14B experiments are intended to validate the effectiveness of this mechanism under controlled and accessible compute settings. Importantly, the additional overhead arises only during training and does not carry over to testing. EAPO should therefore be understood as a training-time-only expert-guided RL framework validated at moderate model sizes.

---

> > ### Author Rebuttal · Reviewer_s4n5 · 2026-04-02
> >
> > The authors have partially addressed my concerns, but I still have the following questions:
> > - Can the authors provide a comparison of the memory usage of EAPO and Self-Exploratory RL?
> > - Regarding the HumanEval and EvalPlus results reported in Appendix D, Table 7, I noticed a more serious issue. The single model originally achieves better performance, but its performance drops when a 7B model is used as the expert. Moreover, when a 32B model that itself does not perform well is used as the expert, the trained 7B model becomes even worse. Could the authors explain this phenomenon?

---

> > > ### Author Response · Authors · 2026-04-02
> > >
> > > **For Q1:**
> > >
> > > Thank you for raising this important point. We would like to clarify that the policy-side training memory of EAPO is essentially comparable to that of Self-Exploratory RL. In both methods, the trainable component is the same policy model, and they share the same key training configuration (**Appendix A, Table 5**), so the memory footprint associated with policy parameters, gradients, optimizer states, and activations remains largely unchanged. Consistent with this, when using DeepSeek-R1-Distill-Qwen-7B as the policy model, the policy training jobs of both Self-Exploratory RL and EAPO are around **3.9–4.0 TB** in average total GPU memory.
> > >
> > > The main difference is that EAPO introduces additional training-time memory for expert serving during rollout. The policy training job requires about **3.9–4.0 TB total** **GPU** **memory on average** and about **4.7 TB at peak** when using policy model of R1-7B, while each expert requires about **660 GB total GPU memory on average** and **670 GB at peak** when using expert of QwQ-32B. Therefore, compared with Self-Exploratory RL, EAPO adds a moderate additional memory overhead of about **14–17%** per expert replica in our setup. Importantly, this overhead comes from serving experts during training, rather than from a larger memory footprint for optimizing the policy model itself.
> > >
> > > Moreover, this overhead is training-only. At test time, EAPO uses only the final policy model and requires no external expert calls, so its inference memory is identical to standard single-model reasoning. Given that EAPO improves over Self-Exploratory RL by about 5 points on average on the math benchmarks (**Table 1**), we believe this training-time memory overhead is a reasonable trade-off for the gain in reasoning quality and stability.
> > >
> > >
> > >
> > > &nbsp;
> > >
> > >
> > >
> > > **For Q2:**
> > >
> > > Thank you for pointing this out. We first clarify that **Appendix D, Table 7** reports Expert-Assisted workflows, **not EAPO**. Thus, the observed changes do not mean that the 7B policy is trained to become worse; rather, they reflect how directly using expert outputs at inference time affects the offline workflow.
> > >
> > > The drop in the 7B+7B+7B setting is consistent with our discussion of self-collaboration (**Appendix C**): when the policy and experts are instantiated from the same 7B model, their responses are highly correlated, so aggregation may amplify redundancy rather than provide complementary information. The degradation in the 32B-expert code setting has a different cause: as discussed in **Appendix D**, QwQ-32B is relatively weak on HumanEval and EvalPlus, so directly injecting its outputs into the offline workflow can introduce distracting or low-quality suggestions on these code-centric tasks.
> > >
> > > This contrast actually highlights a key advantage of EAPO. Unlike offline workflows, EAPO uses expert responses only as training-time assistance, with learnable consultation and progressively reduced reliance, instead of naively relying on expert outputs at test time. As a result, even though the offline workflow degrades on code tasks, EAPO still improves over both the base model and Self-Exploratory RL on HumanEval and EvalPlus in **Table 4**. Therefore, **the contrast between Table 7 and Table 4** further highlights the advantage of EAPO: EAPO is not merely relying on expert outputs, but learning to extract and internalize useful signals from them in a more robust way.

---

### Official Review · Reviewer_WTBP · 2026-02-27

**Soundness:** 2
**Presentation:** 4
**Significance:** 3
**Originality:** 3
**Overall Recommendation:** 4
**Confidence:** 4

**Summary:**

This paper introduces EAPO, a reinforcement learning framework for large language models that enables them to seek assistance from a stronger expert model during training. By incorporating expert feedback into the sampling trajectory, the method encourages the model to gradually internalize expert-level reasoning strategies. Experiments on mathematical reasoning benchmarks demonstrate consistent performance improvements, and the approach also shows generalization to non-mathematical tasks.

**Compliance With Llm Reviewing Policy:**

Affirmed.

**Final Justification:**

Thank you for the detailed and thoughtful rebuttal. The additional experiments and analyses significantly strengthen the paper, particularly in demonstrating robustness across different policy backbones, expert models, and heterogeneous expert settings, as well as providing clearer insights into training dynamics and expert-policy interactions.

Overall, I find the method sound and well-motivated, with improved empirical support and clearer positioning after the rebuttal. While some limitations remain, they are acknowledged and reasonably discussed. Based on these improvements, I update my score from 3 to 4.

**Key Questions For Authors:**

1. The policy model is limited to DeepSeek-R1-Distill-Qwen-7B. Have you tested EAPO on other backbones (e.g., Qwen3 or other recent reasoning LLMs)?
2. The expert pool consists of replicas of the same model. Have you explored heterogeneous experts (e.g., Qwen, GLM, DeepSeek) to increase diversity of reasoning signals? It would be helpful to understand whether diversity itself contributes to the gains.
3. Results in Table 1 shows that distillation performs worse than the base model. Could you clarify why distillation harms performance here?
4. During training, are expert-generated tokens masked from the policy loss?

I would be open to increasing my score if these questions are adequately addressed.

**Limitations:**

No. The paper does not sufficiently discuss practical limitations. In particular, multi-expert querying introduces additional computational overhead, raising concerns about scalability and environmental cost. It would also be helpful to clarify how the method behaves on very hard tasks where the expert frequently fails. A more concrete discussion of these limitations would strengthen the paper.

**Strengths And Weaknesses:**

**Strengths**

1. The paper is clearly written and logically organized, making the motivation, method, and experiments easy to follow.
2. The method addresses an important RL challenge for LLMs, namely reward sparsity and inefficient exploration in long-horizon reasoning tasks.
3. The experimental evaluation is comprehensive, covering both mathematical and non-mathematical benchmarks, which supports the claim of cross-domain generalization.

**Weaknesses**

1. The experiments use a medium sized model QwQ-32B as the expert model, and it is unclear how performance scales with stronger experts e.g. (Qwen3-235B, GLM-4.7, DeepSeek V3.2), and whether the gains would continue or saturate.
2. The claim that EAPO alleviates reward sparsity is not directly supported by empirical analysis (e.g., reward distribution or learning dynamics comparisons with self-exploratory RL).
3. The additional computational cost introduced by expert querying during training is not analyzed, making it difficult to assess the practical trade-offs of the approach.

---

> ### Author Rebuttal · Authors · 2026-03-31
>
> **For W1:**
>
> Thank you for the helpful suggestion. We have added new experiments with stronger expert models while keeping the policy model fixed, using the same Expert-Assisted Workflow setting. The results show that performance improves with stronger experts, indicating that the benefit is not specific to the original QwQ-32B expert, nor do they saturate at that regime. Comparing the results with Llama-8B as the expert in **Table 3** and 32B as the expert in **Table 1** also suggests an expert-size scaling effect, where larger expert models lead to stronger performance.
>
> ||Policy Model|Experts|AIME2024|AIME2025|AIMO2025|Avg.|
> |-|-|-|-|-|-|-|
> |Single Model|R1-7B|--|51.73|37.19|38.68|42.53|
> |Expert-Assisted Workflow|R1-7B|(Qwen3-235B-A22B)*3|60.72|49.58|47.54|52.61|
> |Expert-Assisted Workflow|R1-7B|(GPT-5)*3|61.57|52.96|49.07|54.53|
> |Expert-Assisted Workflow|R1-7B|(DeepSeek_V3.2)*3|61.89|51.24|50.52|54.55|
>
>
>
> &nbsp;
>
>
>
> **For W2:**
>
> Thank you for your thoughtful review. The training curves show that EAPO converges faster throughout training, suggesting that its benefit lies not only in stronger final performance but also in improved training dynamics through more informative optimization signals. In addition, EAPO introduces richer expert guidance beyond terminal rewards, reducing the policy’s reliance on pure exploration and sparse reward signals. We therefore believe it helps mitigate the optimization difficulty in sparse-reward settings. We will add training curve comparison to camera ready.
>
>
>
> &nbsp;
>
>
>
> **For W3 & Limitations:**
>
> The practical training cost of EAPO relative to self-exploration RL is discussed in our response to  **Reviewer CQyp's W3/Q3**.
>
> Regarding hard tasks where the expert may fail, our results suggest that EAPO does not require the expert to be uniformly stronger than the policy model. For example, although QwQ-32B underperforms R1-7B on code benchmarks such as HumanEval and EvalPlus (**Appendix D, Table 7**), EAPO still improves code-task performance in this setting (**Table 4**). This suggests that EAPO can benefit from complementary strengths between the expert and the policy, even when the expert is imperfect in a given domain. In addition, Appendix E demonstrates that when experts are frequently wrong or completely uninformative, performance drops, but the degradation remains bounded.
>
>
>
> &nbsp;
>
>
>
> **For Q1:**
>
> Thank you for your thoughtful review. We have added experiments on Qwen3-8B as an additional policy backbone. The results show the same trend as with R1-7B: EAPO consistently outperforms the base model, the expert-assisted workflow, and self-exploratory RL, with clear gains across benchmarks. This suggests that the effectiveness of EAPO is not specific to R1-7B and R1-14B, but generalizes to other recent reasoning backbones.
>
> |Method|AIME2024|AIME2024|AIME2025|AIME2025|AIMO2025|AIMO2025|Avg.|Avg.|
> |-|-|-|-|-|-|-|-|-|
> ||Pass@32|Var|Pass@32|Var|Pass@32|Var|Pass@32|Var|
> |Base Model|40.84|0.1251|22.62|0.0812|23.57|0.1281|29.01|0.1114|
> |Expert-Assisted Workflow|44.37|0.1960|28.16|0.2127|30.01|0.2224|34.18|0.2104|
> |Self-Exploratory RL|50.21|0.0629|43.52|0.0634|42.59|0.0811|45.44|0.0691|
> |EAPO (Ours)|**53.84**|**0.0610**|**47.03**|**0.0578**|**49.08**|**0.0743**|**49.98**|**0.0644**|
>
>
>
> &nbsp;
>
>
>
> **For Q2:**
>
> Thank you for this thoughtful suggestion. We added an experiment with a heterogeneous expert pool (Qwen3-235B-A22B, GPT-5, and DeepSeek-V3.2) in the same Expert-Assisted Workflow setting. The heterogeneous pool performs slightly better than replicas of a single expert model (**Response to W1**). This suggests that expert diversity provides additional gains, likely due to complementary strengths across models.
>
> ||Policy Model|Experts|AIME2024|AIME2025|AIMO2025|Avg.|
> |-|-|-|-|-|-|-|
> |Expert-Assisted Workflow|R1-7B|Qwen3-235B-A22B + GPT-5 + DeepSeek_V3.2|62.04|52.51|51.08|55.21|
>
>
>
> &nbsp;
>
>
>
> **For Q3:**
>
> Thank you for your question. Our interpretation is that distillation on narrow-domain math Long-CoT data (dapo-math) may transfer the style and surface form of teacher traces more than the student’s underlying standalone reasoning ability. In this setting, this can lead to overfitting to long trace patterns and weaker generalization. Prior work also suggests that long-chain distillation can weaken the learning signal of key reasoning tokens [1].
>
> [1] Chen X, Zhou S, Liang K, et al. Skip-thinking: Chunk-wise chain-of-thought distillation enable smaller language models to reason better and faster[C]//Proceedings of the 2025 Conference on Empirical Methods in Natural Language Processing. 2025: 12153-12168.
>
>
>
> &nbsp;
>
>
>
> **For Q4:**
>
> Yes. Expert-generated tokens are kept as context in the trajectory, but they are masked out from the policy loss. Thus, they influence subsequent policy decisions only through context, rather than through direct token-level supervision.

---

> > ### Author Rebuttal · Reviewer_WTBP · 2026-04-03
> >
> > Thank you for the dedicated response. I will update my recommendation accordingly. However, I still find the explanation incomplete, as prior evidence (e.g., DeepSeek-R1-Distill-Qwen-7B) shows that distillation can significantly improve reasoning performance, suggesting that the degradation observed here may depend on specific data or setup choices and would benefit from further clarification.

---

> > > ### Author Response · Authors · 2026-04-03
> > >
> > > Thank you for the follow-up. Successful cases such as DeepSeek-R1-Distill-Qwen-7B suggest that distillation can improve reasoning performance with sufficient data and careful design.
> > >
> > > In our setting, **for fairness**, our distillation baseline is built from the same DAPO-MATH data and the same QwQ-32B expert used in EAPO, with DeepSeek-R1-Distill-Qwen-7B as the student. We conduct a supplementary analysis of output length. The distilled model produces longer reasoning traces on average, yet achieves lower solve rates than both the base model and EAPO. Under this restricted setup, we believe that narrow-domain Long-CoT SFT appears to transfer the style of teacher traces more readily than standalone problem-solving ability. This interpretation is also **consistent with recent findings**: Luo et al. [1] report that limited long-CoT supervision can lead to Long CoT Degradation, where smaller models generate longer yet less accurate outputs, while Li et al. [2] find that long-CoT or large-teacher CoT primarily changes the student’s reasoning style, rather than consistently improving its generalizable reasoning ability.
> > >
> > > In contrast, EAPO uses expert outputs as contextual guidance during RL optimization rather than direct token-level imitation, which appears more robust in this data-limited setting.
> > >
> > >
> > >
> > > &nbsp;
> > >
> > >
> > >
> > > [1] Luo R, Li J, Huang C, et al. Through the valley: Path to effective long cot training for small language models[C]//Proceedings of the 2025 Conference on Empirical Methods in Natural Language Processing. 2025: 4972-4992.
> > >
> > > [2] Li Y, Yue X, Xu Z, et al. Small models struggle to learn from strong reasoners[C]//Findings of the Association for Computational Linguistics: ACL 2025. 2025: 25366-25394.

---

### Official Review · Reviewer_jL3h · 2026-03-13

**Soundness:** 2
**Presentation:** 3
**Significance:** 2
**Originality:** 2
**Overall Recommendation:** 3
**Confidence:** 2

**Summary:**

This paper proposes EAPO, an RL framework for reasoning models in which the policy is trained not only to generate reasoning steps, but also to decide when to invoke external experts during training. The main modeling choice is to treat expert consultation as an explicit learnable action $\alpha_t$ jointly optimized with the policy’s reasoning behavior under the same end-to-end reward. At test time, expert access is disabled, so the goal is to internalize useful expert behavior during training while preserving autonomous inference at evaluation time. The method allows policy to either solve problems independently, consult multiple experts on the whole problem, or decompose a problem into subproblems and query experts on each piece. The training design gradually reduces access to expert responses by annealing the effective acceptance rate of expert feedback over training. Across these experiments, the paper argues that expert consultation during RL improves both performance and stability and that some of the learned reasoning behavior transfers beyond the training domain.

**Compliance With Llm Reviewing Policy:**

Affirmed.

**Final Justification:**

I thank the authors for the detailed rebuttal. It addressed several of my earlier concerns in a meaningful way: the added consultation-schedule ablation, the stronger RL baseline comparison, the clearer discussion of expert-quality dependence, and the new training-cost table all improve the paper. I also appreciate the authors’ willingness to weaken the internalization claim and to frame section 2.2 as an empirical hypothesis rather than a mechanistic explanation. These revisions improve both the clarity of the paper and the calibration of the claims.

That said, I am keeping my final recommendation at 3 (Weak Reject). The main remaining issue is that the core internalization story is still supported indirectly rather than causally: reduced consultation during training and no-expert test-time gains are consistent with internalization, but they do not establish that the policy has actually absorbed expert reasoning rather than benefiting from other training effects. The new cost comparison is helpful, but it also seems to confirm a nontrivial training overhead relative to self-exploratory RL. I think the paper has real strengths in idea, empirical results, and organization, and the rebuttal improved my view of it, but I still find the remaining soundness/significance concerns enough to keep it just below the accept bar.

**Key Questions For Authors:**

- How sensitive is EAPO to the consultation annealing schedule?
- How much of the gain comes from stronger exploration versus simply exposing the policy to extra high-quality reasoning tokens from the expert?
- What is the actual training-time cost of EAPO relative to self-exploratory RL and offline workflows?

**Limitations:**

- One limitation is that the “internalization” claim is only indirectly validated. The consultation-rate plots are suggestive, but they do not fully prove that the policy has absorbed expert reasoning strategies in a causal sense.
- Are there direct limitations section covering expert quality dependence? Table 2 and Table 3 already suggest that performance depends materially on the capacity and heterogeneity of experts. If expert guidance is weak, redundant, or poorly matched to the policy, the method can lose much of its value which is important to see.

**Strengths And Weaknesses:**

### Strengths
- EAPO does not just add expert text to prompts - it formalizes expert consultation as a learnable action inside the trajectory, with $\pi_{\alpha}^\tau$ generating reasoning chains and $\pi_{\alpha}^\theta$ deciding whether to consult experts which is interesting
- There are useful ablations on the expert side ie: parallel expert querying with a 32B expert beats sequential EAPO with the same 32B expert and parallel expert querying with a 14 B expert
- The cross-domain generalization results make the paper more interesting than a math-only optimization strategy since EAPO still improves over both the base model and self-exploratory RL on non-math benchmarks
- paper is organized logically and is easy to follow at a high level especially in terms of the figures

### Weaknesses
- The consultation annealing rule is under-justified: the probability of writing expert responses into history is set to $\rho_s = s^{-1}$ yet I did not see an ablation or explanation on  schedule itself, nor a comparison to slower decay, constant access, or learned gating schedules
- The paper’s “mechanistic intuition” is much more heuristic than mechanistic. Section 2.2 argues that expert assistance reduces sparse rewards, lowers uncertainty, and decreases gradient variance, but these are not presented via a formal analysis. The section explicitly relies on assumptions such as “the expert knowledge is valid for the current task” and “the policy model can extract useful features from the assistance,” but these assumptions are not really stress-tested. I think the claims in this section should be presented more modestly as hypotheses supported by experiments rather than as mechanism.
- The baseline suite is solid but still seems incomplete for a paper making broad claims about RL reasoning optimization. The main online comparison is essentially against self-exploratory RL, plus offline collaboration and distillation baselines. Comparisons against stronger or more recent RL variants using intermediate supervision or alternative exploration schemes, rather than mostly showing that “RL + expert assistance” beats “RL without expert assistance,” which is not surprising.
- Several awkward phrasing: for example “the agent is allowed to consult experts freely without in the early stages of training,” “the output answer $\hat{y}$ is extracted from the last reasoning step,” and other minor grammatical issues
- Consultation policies, expert-guided learning, imitation/distillation from stronger agents, and hierarchical control are studied in literature, and the related work section itself acknowledges many of them. The novelty here comes from the specific combination and the training-time-only use of experts, not from a fundamentally new learning principle which should be emphasized.

---

> ### Author Rebuttal · Authors · 2026-03-31
>
> **For W1 & Q1:**
>
> Thank you for your review. We have addressed this concern in our response to **Reviewer CQyp’s W2/Q2** by adding an ablation over consultation schedules. For learned gating, we intentionally adopt a simple hand-designed schedule to keep the mechanism interpretable. Learned gating schedules are a promising future direction, but they introduce an additional learnable component beyond the scope of this paper.
>
> ||AIME2024|AIME2025|AIMO2025|Avg.|
> |-|-|-|-|-|
> |$$\rho _s=0.02$$(Budget-Matched)|67.92|55.13|55.89|59.65|
> |$$\rho _s=s^{-1/2}$$(Slow Delay)|68.53|57.41|62.53|62.82|
> |$$\rho _s=s^{-2}$$ (Faster Delay)|68.04|55.21|56.03|59.76|
> |$$\rho _s=s^{-1}$$ (Ours)|**70.03**|**58.01**|**64.17**|**64.07**|
>
>
>
> &nbsp;
>
>
>
> **For W2 & L2:**
>
> We thank the reviewer for this helpful comment. We would like to clarify that Section 2.2 is intentionally named as **“Mechanistic Intuition”**, rather than a formal theoretical analysis.
>
> At the same time, they are not entirely untested, some experiments explored the expert quality dependence. In **Appendix E/Table 8**, we further design two robustness experiments. The results show that low-quality or non-informative expert signals do hurt performance, but the degradation is bounded, while the standard EAPO setting remains consistently better. In **Appendix D/Table 7**, QwQ-32B is weaker than R1-7B on HumanEval and EvalPlus. Yet in **Table 4**, EAPO still improves on both benchmarks, which suggests that the benefit can come from complementary capability distributions between policy and expert, rather than raw expert strength alone.
>
> In addition, the observed changes in reasoning patterns on EAPO training suggest that the model learns useful information beyond self-exploratory RL alone, and that this additional information is beneficial for optimization.
>
>
>
> &nbsp;
>
>
>
> **For W3:**
>
> We thank the reviewer for the suggestion and agree that stronger RL-style baselines are valuable. In addition to self-exploratory RL and other baselines shown in Table 1, we compared EAPO with several post-training / RL baselines built on the same R1-7B backbone, including AceMath-RL-Nemotron-7B, AReaL-boba-RL-7B, and OREAL-DSR1-Distill-Qwen-7B. EAPO still performs best on both AIME2024 and AIME2025. This added comparison strengthens the empirical support beyond the original “with vs. without expert assistance” comparison.
>
> |Method|AIME2024|AIME2025|
> |-|-|-|
> |AceMath-RL-Nemotron-7B|69.0|53.6|
> |AReaL-boba-RL-7B|61.9|48.3|
> |OREAL-DSR1-Distill-Qwen-7B|50.0|40.0 (Part1)|
> |Self-Exploratory RL|67.16|55.09|
> |EAPO (Ours)|**70.03**|**58.01**|
>
>
>
> &nbsp;
>
>
>
> **For W4:**
>
> Thank you for your comments on the wording and grammar. We will further polish and revise them in the camera-ready version.
>
>
>
> &nbsp;
>
>
>
> **For W5:**
>
> Thank you for your comments. We would like to clarify that the contribution of this work does not lie in a specific combination of expert guidance strategies. Instead, our method should be viewed as a **training-time knowledge internalization mechanism**, where expert guidance is absorbed into the student model’s policy rather than remaining as external assistance or simple imitation. This allows the student to reason more effectively on its own at test time.
>
> Moreover, within RL training, expert-provided signals offer richer supervision than terminal rewards alone, helping mitigate sparse-reward challenge while improving training efficiency.
>
>
>
> &nbsp;
>
>
>
> **For Q2:**
>
> We agree that part of the gain may come from the additional high-quality reasoning tokens provided by the expert. However, our results suggest that this factor alone does not explain the improvement. If simple exposure to expert reasoning were sufficient, then distillation baselines and the offline expert-assisted workflow should be much closer to EAPO; empirically, both are clearly weaker than EAPO in **Table 1.**
>
> This suggests that the gain comes not only from expert-generated tokens, but also from how those signals are integrated into on-policy RL training, leading to improved exploration better credit assignment.
>
>
>
> &nbsp;
>
>
>
> **For Q3:**
>
> Offline workflows do not introduce additional training cost, since they only require the construction of the workflow itself. Compared with self-exploratory RL, the practical training cost of EAPO is discussed in our response to **Reviewer CQyp's W3/Q3**.
>
>
>
> &nbsp;
>
>
>
> **For L1:**
>
> We agree that the current validation of “internalization” is indirect. Our claim is therefore not that the consultation-rate plots alone constitute causal proof of internalization, but rather that the overall evidence is consistent with this interpretation. Specifically, the final policy is evaluated without expert assistance at test time, yet still outperforms self-exploratory RL, suggesting that it has internalized useful reasoning patterns induced by expert-guided training. The reduced consultation rate should therefore be viewed as supporting evidence.

---

> > ### Author Rebuttal · Reviewer_jL3h · 2026-04-04
> >
> > Thank you for the detailed rebuttal. The added consultation-schedule ablation, the stronger RL baseline comparison, and the additional discussion of expert-quality dependence materially improve the paper and resolve several of my original concerns.
> >
> > The main remaining issue for me is that the “internalization” claim is still supported indirectly rather than causally. I agree that the reduced consultation rate during training and the no-expert test-time evaluation are consistent with internalization, but they do not fully establish that the policy has absorbed expert reasoning rather than benefiting from other training effects.
> >
> > Relatedly, I still think Section 2.2 should be framed strictly as empirical hypothesis rather than as a mechanistic explanation.
> >
> > My main remaining practical question is about cost. The rebuttal points to another response for this, but I still think the paper would benefit from a concrete training-cost comparison against self-exploratory RL and the offline expert-assisted workflows, ideally in terms of expert calls, generated tokens, and wall-clock. I am keeping my score unchanged.

---

> > > ### Author Response · Authors · 2026-04-04
> > >
> > > Thank you for the thoughtful follow-up.
> > >
> > > **First**, regarding “internalization”, we agree that our current evidence is **behavioral** rather than causal. In the revision, we will therefore **weaken the wording** throughout the paper and present internalization as an **empirically supported interpretation** rather than a proven mechanism.
> > >
> > >
> > >
> > > &nbsp;
> > >
> > >
> > >
> > > **Second**, we agree that Section 2.2 should be framed strictly as **empirical hypothesis**. In the revision, we will rewrite that section accordingly and replace mechanistic wording with more conservative language.
> > >
> > >
> > >
> > > &nbsp;
> > >
> > >
> > >
> > > **Third**, we now provide a **cost comparison in the table below**. The main takeaway is that, relative to Self-Exploratory RL, EAPO introduces a moderate additional training-time overhead, but this overhead is not primarily explained by substantially longer outputs; rather, it mainly comes from expert interaction during rollout. In return, EAPO improves over Self-Exploratory RL by about **5 points** on average on the math benchmarks (**Table 1** in Section 3.2). Importantly, this extra cost is **training-only**: at inference time, the final EAPO model uses only the policy model and requires no external expert calls, so deployment remains identical to standard single-model reasoning.
> > >
> > >
> > >
> > > |                          | Expert Calls  (Training Phase) | Expert Calls  (Inference Phase) | Generated Tokens (Training Phase) | Wall-Clock (Training Phase) |
> > > | ------------------------ | ------------------------------ | ------------------------------- | --------------------------------- | --------------------------- |
> > > | Expert-Assisted Workflow | --                             | 210 / 70 rollouts               | --                                | --                          |
> > > | Self-Exploratory RL      | --                             | --                              | 45M                               | 70 hours * 64 H20 GPUs           |
> > > | EAPO                     | 118 / 6000 rollouts            | --                              | 46.8M                             | 100 hours * 64 H20 GPUs          |
> > >
> > >
> > > &nbsp;
> > >
> > > This also clarifies the distinction from Expert-Assisted Workflows. Such workflows do not incur training cost, but they do rely on **continued expert interaction** at inference time. By contrast, EAPO shifts part of that burden into training, with the goal of **improving the policy itself** while preserving **test-time independence**.
> > >
> > >
> > > &nbsp;
> > >
> > > We appreciate this suggestion because it helps us make the claims of the paper more rigorous and precise. If these clarifications and revisions address your remaining concerns, we would sincerely appreciate your reconsideration of the score.

---

### Official Review · Reviewer_CQyp · 2026-03-13

**Soundness:** 3
**Presentation:** 3
**Significance:** 3
**Originality:** 3
**Overall Recommendation:** 4
**Confidence:** 5

**Summary:**

This paper proposes **EAPO (Expert-Assisted Policy Optimization)**, a reinforcement learning framework for LLM reasoning that treats expert consultation as a learnable action. During training, the policy can explicitly invoke external experts at critical steps, incorporate their guidance into its reasoning trajectory, and gradually reduce reliance as competence improves. The key innovation lies in framing consultation as a decision variable optimized end-to-end within the RL objective, rather than using experts as static teachers or fixed heuristics.

Empirically, EAPO achieves strong results on mathematical reasoning benchmarks (AIME 2024/2025, AIMO 2025), outperforming self-exploratory RL (Pass@32: **64.07** vs. **59.16**) with lower variance (**0.0643** vs. **0.0727**). The paper includes comprehensive ablations on expert size, query parallelism, homogeneous vs. heterogeneous expert pools, and cross-domain transfer to non-mathematical tasks. A systematic analysis of consultation dynamics shows that expert invocation decreases throughout training while performance remains stable, supporting the claim of capability internalization.

**Compliance With Llm Reviewing Policy:**

Affirmed.

**Final Justification:**

My concerns are addressed. I will keep my postive score.

**Key Questions For Authors:**

### Q1. Heterogeneous Non-EAPO Baseline

Could you add a "7B policy + 32B experts" offline workflow baseline (without RL training) to Table 1? This would isolate gains from the RL mechanism versus gains from expert quality. If the offline heterogeneous baseline is close to EAPO, this would challenge the claim that the RL formulation is essential.

### Q2. Annealing Necessity

Please provide an ablation comparing the annealed consultation schedule ($\rho_s = s^{-1}$) against a fixed consultation rate. Does annealing actually improve final performance, or does it merely reduce expert usage?

### Q3. Compute Cost Analysis

What is the total training cost (GPU-hours) of EAPO compared to self-exploratory RL? Given that 32B Self Collaboration (72.03) outperforms 7B EAPO (64.07), under what conditions is the training overhead justified?

**Strengths And Weaknesses:**

## Strengths
1. The paper identifies a genuine gap in current approaches: sparse verifiable rewards make exploration inefficient, yet test-time scaling methods don't improve the underlying policy. The solution—learnable consultation that fades over time—is conceptually clean and practically appealing.

2. EAPO consistently outperforms the strongest online RL baseline across all three mathematical benchmarks. The gains are substantial  and accompanied by reduced variance, suggesting more stable optimization rather than lucky outliers.

3. The paper goes beyond the main result with valuable analyses:

- **Parallel vs. sequential querying** shows information coverage matters
- **Heterogeneous vs. homogeneous experts** (Table 3) reveals complementary capability distributions are crucial—an important empirical finding


### 4. Cross-Domain Validation

Evaluation on seven non-mathematical benchmarks (Table 4) strengthens the claim that EAPO induces general reasoning behaviors rather than math-specific hacks.

---

## Weaknesses

1. The main comparison confounds expert heterogeneity with the RL mechanism. While Table 1 includes "Self Collaboration" (homogeneous 7B+7B+7B) and Table 3 shows heterogeneous experts help within EAPO, there is **no heterogeneous non-EAPO baseline** (e.g., 7B policy with offline access to 32B experts without RL training). This makes it difficult to attribute gains to the proposed RL framework versus simply using stronger, diverse collaborators.

> **Critical Observation:** Appendix C (Table 6) shows **32B Self Collaboration achieves 72.03**—higher than **7B EAPO (64.07)**—raising questions about whether the gains come from the method or merely from access to better experts.

2. The paper invokes "information gain," "sparse reward alleviation," and "capability internalization" as mechanisms, but the evidence is mostly suggestive rather than causal. The decreasing call rate is consistent with internalization, but could also reflect the forced annealing schedule ($\rho_s = s^{-1}$). A stronger test would compare annealed vs. fixed-rate consultation.

3. Training with three 32B expert replicas is expensive. The paper shows that larger experts and parallel queries improve results, but does not quantify the cost-performance trade-off. **When is EAPO preferable to simply using a larger model or test-time scaling?**

---

> ### Author Rebuttal · Authors · 2026-03-31
>
> **For W1 & Q1:**
>
> Thank you for your thoughtful review. The requested heterogeneous non-EAPO baseline is already included in **Table 1** as Expert-Assisted Workflow, where the 7B policy collaborates with stronger 32B experts without RL training. Therefore, the comparison with EAPO already isolates the effect of the proposed RL framework, rather than simply reflecting access to stronger collaborators.
>
> Moreover, the strong performance of 32B Self-Collaboration (Appendix C) does not weaken our claim. It primarily reflects the much higher intrinsic capability of the 32B model itself. Our goal is to show that expert interaction during training can improve the standalone reasoning ability of the 7B policy model. In this respect, 7B EAPO uses no external experts at inference time, whereas 32B Self-Collaboration still depends on multiple strong external models at test time. Thus, the core value of EAPO lies in capability transfer and internalization, not merely expert-assisted test-time performance.
>
>
>
> &nbsp;
>
>
>
> **For W2 & Q2:**
>
> Thank you for this helpful suggestion. We have added the requested ablation comparing the annealed consultation schedule with a budget-matched fixed-rate baseline, together with slower and faster decay variants. Across all three benchmarks, the annealed schedule achieves the best final performance, outperforming both the fixed-rate baseline and the alternative decay schedules.
>
> This comparison suggests that annealing does more than simply reduce expert usage. Because the fixed-rate baseline is matched in total consultation budget, its weaker performance indicates that the gain is not explained by fewer/more expert calls alone. Rather, the results are consistent with our intended interpretation: stronger expert access is most helpful in the early, unstable stage of training, while gradual withdrawal leads to a better final standalone policy than either keeping the consultation rate fixed or decaying it too slowly/too aggressively.
>
> In addition, we acknowledge that some of our mechanistic claims would benefit from more cautious framing.
>
> |                                    | AIME2024  | AIME2025  | AIMO2025  | Avg.      |
> | ---------------------------------- | --------- | --------- | --------- | --------- |
> | $$\rho _s=0.02$$  (Budget-Matched) | 67.92     | 55.13     | 55.89     | 59.65     |
> | $$\rho _s=s^{-1/2}$$  (Slow Delay) | 68.53     | 57.41     | 62.53     | 62.82     |
> | $$\rho _s=s^{-2}$$ (Faster Delay)  | 68.04     | 55.21     | 56.03     | 59.76     |
> | $$\rho _s=s^{-1}$$ (Ours)          | **70.03** | **58.01** | **64.17** | **64.07** |
>
>
>
> &nbsp;
>
>
>
> **For W3 & Q3:**
>
> Thank you for raising this important point. In our setup, self-exploratory RL takes about 70 hours on 8×8 H20 GPUs, whereas EAPO requires approximately 100 hours of training on 8×8 H20 GPUs, corresponding to about 1.4× the training time of self-exploratory RL, mainly due to expert queries during rollout. This additional cost is incurred only during training. At test time, the final EAPO model uses only the 7B policy model and requires no external expert calls, so its inference procedure remains identical to standard single-model reasoning.
>
> We would like to clarify that the goal of EAPO is not to outperform a much larger multi-model system under expert-assisted inference. Rather, EAPO is designed for the setting where one wants to improve the capability of the policy model itself, while still keeping evaluation and deployment in a single-model form without constructing a multi-agent collaboration system at test time. Under this objective, the extra training overhead is meaningful because it transfers useful reasoning behaviors from stronger experts into the policy and removes the need for continued expert participation after training.
>
> By contrast, results such as 32B Self-Collaboration correspond to a different operating setting: they rely on a much stronger backbone together with continued multi-model coordination during inference. Such methods may achieve higher absolute accuracy, but they do not address the same goal as EAPO, namely training-time capability transfer with test-time independence.

---

> > ### Author Rebuttal · Reviewer_CQyp · 2026-04-02
> >
> > My concerns are addressed. I will keep my postive score.

---

> > > ### Author Response · Authors · 2026-04-03
> > >
> > > Thank you again for your thoughtful review and for confirming that our rebuttal has addressed your concerns. We sincerely appreciate your time, careful consideration, and positive assessment of our work.
> > >
> > > We are especially appreciative that the additional clarifications, the new annealing ablation, and the compute-cost analysis were helpful in resolving the main questions raised in your original review. We believe these additions directly address the issues that initially limited the paper’s evaluation.
> > >
> > > If, in light of these updates, you feel that the original concerns no longer materially affect your overall assessment of the paper, we would **be very grateful if you would consider improving the score** to reflect your revised view. We **fully respect your judgment either way**, and we truly appreciate your constructive feedback throughout the review process.

---

### Decision · Program_Chairs · 2026-04-30

**Decision:**

Accept (regular)

**Comment:**

The authors introduce EAPO, a RL framework that consults external experts during training. It incorporates the external guidance during training and gradually reduces the reliance on them as the training converges. The method leads to consistent gains on reasoning benchmarks (AIME 2024/2025, AIMO 2025). It also generalizes to non-mathmatical reasoning datasets such as HotpotQA and MMLU. Systematic comparisons have been shown for different settings. The detailed analysis is appreciated. While there are concerns about framing of the paper from reviewer jL3h and reviewer CQyp. Reviewer s4n5 had concerns on the memory usage, which the authors clarified during the rebuttal. It seems all reviewers have acknowledged the amount of work done by the authors. The reviewers leaning towards acceptance have a higher confidence and the reviewers leaning towards rejection have a lower confidence.